# Kinetic properties of optogenetic site-specific DNA recombination by LiCre-*loxP*

Alice Dufour[1,*], Hélène Duplus-Bottin[1,*], Thomas Boukéké-Lesplulier[1], Eliane Casassa[1], Gérard Triqueneaux[1], Léo Tarbouriech[1], Camille Darthenay-Kiennemann[1], Agnès Dumont[1], Catherine Moali[2], Franck Vittoz[3], Daniel Jost[1,§] and Gaël Yvert[1,‡,§]

## ABSTRACT

Advances in optogenetics now allow specific modifications to the DNA of live cells with light. However, successfully using these technologies requires knowing their properties in terms of sensitivity, efficiency, kinetics and mechanism. We previously developed an optogenetic tool made of a single chimeric protein called LiCre that enables the induction of specific changes in the genome with blue light via DNA recombination between *loxP* sites (Duplus-Bottin et al., 2021). Here, we used *in vitro* and *in vivo* experiments combined with kinetic modeling to provide a deeper characterization of the photoactivated LiCre-*loxP* recombination reaction. We find that LiCre binds DNA with high affinity in the absence of a light stimulus and that this binding is cooperative, although not as much as for the Cre recombinase from which LiCre was derived. In yeast, the addition of riboflavin to the culture medium had no effect on LiCre's efficiency, even when cells over-expressed riboflavin kinase, suggesting that the abundance of the flavin mononucleotide cofactor is not limiting for the reaction. However, LiCre's efficiency in yeast gradually increased when raising the temperature from 20°C to 37°C. The recombination kinetics observed in live cells are best explained by a model where the photoactivation of two or more DNA-bound LiCre units (happening in seconds) can produce (in several minutes) a functional recombination synapse. This model was able to capture the effect of a point mutation altering LiCre's light cycle. This deeper understanding of the LiCre-*loxP* system provides additional knowledge for designing experiments where specific genetic changes are induced in live cells with light.

KEY WORDS: Optogenetics, DNA editing, Recombinase, Cre-*loxP*, Genetics

## INTRODUCTION

Experimental biologists usually make discoveries by perturbing living organisms and observing the consequences of these perturbations. Precision in these manipulations is often crucial because biological systems are spatially complex and highly dynamic. Optogenetics offers the possibility to alter cellular activities with light and provides unprecedented spatiotemporal resolution. For this reason, a wide set of light-inducible biotechnologies has been developed in recent years. These tools include photoactivatable ion channels that trigger neuronal activities when illuminated (see Altahini et al., 2024, for a review on neuroscience applications), photo-controlled kinases activating signaling pathways and systems controlling protein localization, protein-protein interactions, gene expression or DNA editing (see Armbruster et al., 2024, for a review on molecular tools). Many of these systems are genetically encoded, thereby alleviating the need for chemicals. They are all based on conformational changes of a macromolecule, leading to its activation or inactivation in response to light. Yet each tool has its own specificity in terms of light sensitivity, efficiency and response dynamics; optimal usage of a given tool therefore requires quantitative knowledge of its biochemical properties.

We recently developed an optogenetic site-specific DNA recombinase called LiCre (Duplus-Bottin et al., 2021). As for the Cre recombinase from which it derives, LiCre recognizes 34-bp DNA sequences called *loxP* and catalyzes DNA recombination between two such sites. This allows experimentalists to induce specific changes in the genome of live cells in real time with blue light. For example, if a gene is flanked by two *loxP* sites in LiCre-expressing cells, shining light on these cells can delete the gene. Compared to other photoactivatable recombinases, which are based on two protein halves (splitCre) that complement in response to light (Hochrein et al., 2018; Kawano et al., 2016; Kuwasaki et al., 2022; Taslimi et al., 2016), LiCre is unique as it consists of a single chimeric protein comprising the light oxygen voltage domain from *Avena sativa* (asLOV2) fused to a mutant version of Cre. It was designed such that the unfolding of a critical α helix in response to light enables the assembly of a functional recombination synapse (Duplus-Bottin et al., 2021). The mode of action of LiCre, represented in Fig. 1A, is speculated from the abundant knowledge previously acquired on the Cre-*loxP* synapse (Van Duyne, 2015) and the asLOV2 photo switch (Zayner et al., 2012). Formation of the Cre-*loxP* complex involves the cooperative binding of two Cre units on each *loxP* site and the subsequent assembly of four Cre proteins complexed with their DNA substrates (Van Duyne, 2015). The resulting intasome is stabilized by interactions between adjacent protomers: on one side of the DNA substrate, the four C-terminal domains are locked together in a cyclic manner, each unit hosting the αN helix of its neighbor in a nest; on the other side of the DNA (depicted in Fig. 1A for LiCre),

[1]Laboratory of Biology and Modeling of the Cell, Ecole Normale Supérieure de Lyon, CNRS, UMR 5239, Inserm, U1293, Université Claude Bernard Lyon 1, 46 allee d'Italie F-69364 Lyon, France. [2]Tissue Biology and Therapeutic Engineering Laboratory, Université Claude Bernard Lyon 1, CNRS, UMR 5305, 7 Passage du Vercors F-69367 Lyon, France. [3]Laboratory of Physics, Ecole Normale Superieure de Lyon, CNRS, UMR 5672, Universite Claude Bernard Lyon 1, 46 allee d'Italie F-69364 Lyon, France.
*These authors contributed equally to this work.
‡Lead contact: gael.yvert@ens-lyon.fr

§Authors for correspondence (gael.yvert@ens-lyon.fr; daniel.jost@ens-lyon.fr)

H.D.-B., 0000-0003-2029-5646; D.J., 0000-0002-9877-6864; G.Y., 0000-0003-1955-4786

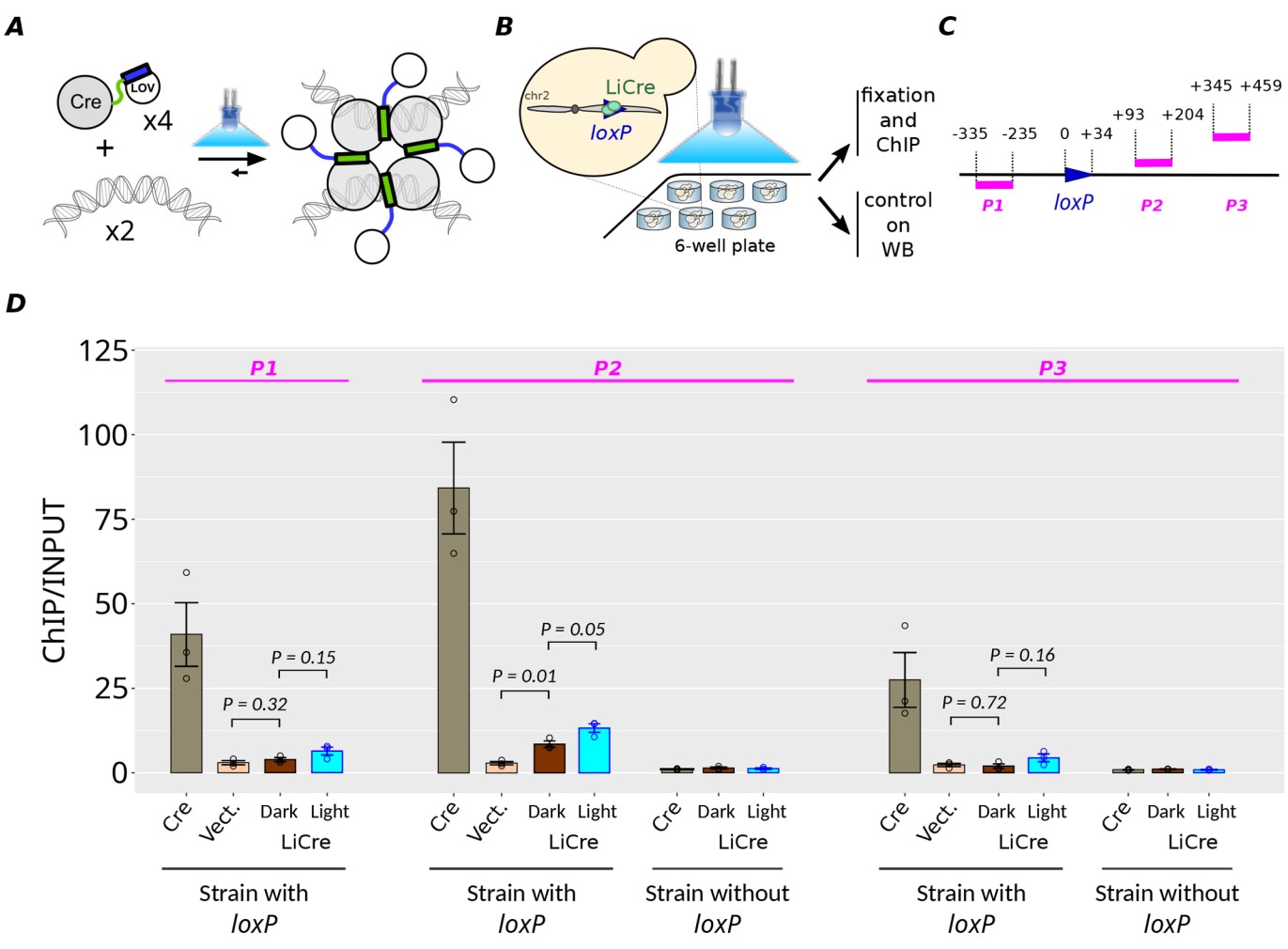

**Fig. 1. Analysis of LiCre:DNA binding *in vivo*.** (A) Hypothetical principle of LiCre photoactivation. The chimeric α helix is presented as a rectangle; in blue when folded as the Jα helix of LOV2 and in green when folded as the αA helix of Cre. Modified from Duplus-Bottin et al. (2021). (B) Chromatin immunoprecipitation (ChIP) experimental setup. (C) Position of qPCR-amplified regions *P1*, *P2* and *P3* relative to *loxP* on the synthetic construct integrated in chr2 of strain GY2416. (D) ChIP-qPCR quantifications. Every dot corresponds to one of *n*=3 biological replicates [independent culture inoculated from an independent transformant of the Cre-(V5)$_{9x}$ (pGY820) or LiCre-(V5)$_{9x}$ (pGY605) expression plasmid]. Values correspond to enrichment by ChIP (relative to its input), normalized by the signals obtained on three unrelated loci of the genome (see Materials and Methods). *P*-values: two-tailed Welch *t*-test. Error bars: mean±s.e.m. Strain with *loxP*: GY2416. Strain without *loxP*: GY2758, which contains synthetic sites *P2* and *P3* but not *loxP* and *P1*. This specific experiment was performed once in the laboratory. An additional, equivalent and independent experiment is shown in Fig. S1.

N-terminal domains also form a cycle of binding interfaces but in the reverse orientation (Guo et al., 1997). This way, each protomer is stabilized by its two neighbors. For LiCre, the strength of interactions between C-terminal domains was reduced by mutating residues E340 and D341, which, via their attraction to arginines R192 and R139, stabilize the association of the αN helix with the adjacent unit; the association of N-terminal domains was perturbed by truncating the αA helix interacting with the neighboring unit; and photoactivation was obtained after designing a chimeric α helix comprising Jα from asLOV2 and the remaining part of αA from Cre (Duplus-Bottin et al., 2021).

Although LiCre outperformed split-based systems in terms of speed and efficiency (Duplus-Bottin et al., 2021), two essential questions remain on its properties. First, it is unclear if its association with *loxP* occurs before or after its photoactivation. Second, the observed dependency of its efficiency on the dynamics of the light stimulus is not explained. Here, we address these questions by combining *in vitro* and *in vivo* experiments with kinetic modeling. We found that LiCre units bind to DNA with high affinity even if they

are not photoactivated; that they do so in a cooperative manner although this cooperativity is weaker than for Cre; that temperature greatly affects LiCre efficiency; and that the kinetics of recombination in live cells are best explained by a model where the photoactivation of two or more DNA-bound LiCre units happens in seconds and enables the formation, in several minutes, of a functional recombination synapse; and we identify a point mutation in LiCre that modifies its light cycle.

## RESULTS
### LiCre binds its target DNA in the absence of photo-stimulation

LiCre photoactivation is based on a conformational change that enables protein-protein interactions, assembling the recombinogenic intasome (Duplus-Bottin et al., 2021). We previously proposed two possible models for this activation. In the first model, LiCre monomers are unable to bind DNA *loxP* sites in their 'dark' – inactive – state, and the photoinduced conformational change restores this ability. In the alternative model, LiCre binds to *loxP* sites even in

the dark, and the conformational change of photo-stimulated DNA-bound LiCre molecules enables the assembly of the recombination synapse. To distinguish between these two models, we evaluated the ability of LiCre to bind DNA in conditions where it was not photoactivated.

We first tested this DNA-binding ability *in vivo* by using chromatin immunoprecipitation (ChIP). We constructed yeast strains harboring a single *loxP* site in the genome and expressing LiCre or Cre tagged with repeated V5 epitopes (Fig. 1B). We cultured cells overnight and then either let them in dark conditions or, for LiCre-V5 samples, illuminated them in conditions known to trigger LiCre activity (1 s of blue light 450 nm, 35 mW/cm², applied every 10 s for a total duration of 1 h) (Duplus-Bottin et al., 2021). We quantified ChIP by quantitative PCR (qPCR) at three positions near the *loxP* site (Fig. 1C) and, for normalization, at three unrelated positions on the genome (*ISW1* on chrII, *HIS5* on chrIX and *AHA1* on chrIV). As shown in Fig. 1D and Fig. S1, we observed a strong ChIP signal for all three probes of the *loxP* locus for Cre, consistent with the known high affinity of Cre for its *loxP* target (Van Duyne, 2015). Regarding LiCre, we observed a much weaker ChIP signal, which reproducibly reached statistical significance only for the probe located very close to *loxP* (probe P2 in Fig. 1C,D). As expected, no ChIP signal was visible when using a control strain lacking the *loxP* site. There are several possible explanations for the difference in ChIP signals between LiCre and Cre. First, given that ChIP is a crude assay where fragmented particles of cross-linked material are pulled down using antibodies, its output signal could vary if the two proteins differ in their subcellular distribution (e.g. nuclear/cytoplasmic ratio), in the way cross-linking affects the accessibility and/or affinity of the antibody (conformation, local molecular crowding), or in their degree of nonspecific interactions with cellular factors. Alternatively, expression levels of the two proteins may differ.

However, western blots (WBs) showed that, in denaturing conditions, the detection of Cre-V5 by the anti-V5 antibody was not stronger than the detection of LiCre-V5 (Fig. S1b). This rules out a higher expression level for Cre-V5 than LiCre-V5 but does not exclude the possibility of different immuno-affinities in pulled-down conditions. Comparing the DNA affinities of LiCre and Cre therefore requires complementary experiments (provided below). Very importantly, illumination prior to ChIP had only a very small effect on LiCre ChIP signals. This improvement was marginally significant in one experiment (Fig. 1D, probe P2) and not significant at all in another independent experiment (Fig. S1). We conclude that, *in vivo*, LiCre weakly binds *loxP*-containing chromatin in the dark and that this binding may be slightly improved in light conditions.

To better quantify LiCre:DNA binding, we analyzed this affinity *in vitro* by surface plasmon resonance (SPR). This method offers high sensitivity to monitor macromolecular associations and dissociations; it was previously used to characterize the binding affinity of Cre to *loxP* (Rüfer et al., 2002). We produced and purified recombinant Cre and LiCre proteins from *Escherichia coli* (Fig. S2) and injected these proteins into microfluidic chips containing immobilized biotinylated DNA substrates (see Materials and Methods). We used DNA substrates containing either a full *loxP* site that can associate with two units of Cre or LiCre or a half site allowing the binding of only one unit. We also used a randomized DNA sequence to quantify nonspecific binding. We did not apply any blue-light illumination to the samples, neither before injection of the protein solution in the chip nor during acquisitions.

As shown in Fig. 2A, Cre association with the full *loxP* site was very rapid and its dissociation was extremely slow. This observation is in perfect agreement with previous experiments (Rüfer et al., 2002) based on the same DNA substrates as here. For LiCre, association with the full *loxP* site was slightly slower than for Cre but still very

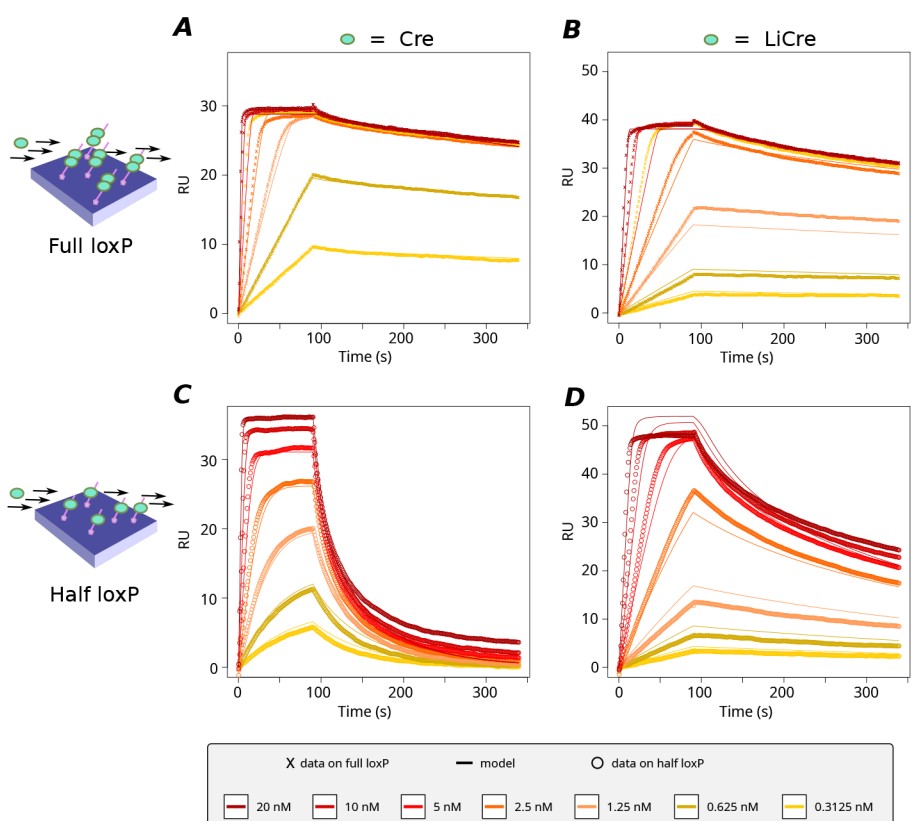

**Fig. 2. Analysis of LiCre:DNA interaction *in vitro* by surface plasmon resonance (SPR).** (A-D) Net sensorgrams showing Cre (A,C) or LiCre (B,D) interaction with full (A,B) or half (C,D) *loxP* sites. At time zero, recombinant proteins were injected at the indicated concentrations over a BIACORE sensor chip containing immobilized *loxP* sites (A,B), *loxP* half sites (C,D) or unrelated DNA as a negative control blank. After 90 s, the sensor chip was washed by injecting running buffer lacking the protein (dissociation step). Net sensorgrams were generated from the raw data by subtracting the negative blank control to correct injection-related fluctuations. Dots, observed values. Lines, predicted values from models (see text and Materials and Methods). Model parameters for Cre (A,C) were $\alpha$=0.87, $RUmax_h$=38.2, $RUmax_f$=28.8, $k_f$=0.51, $k_1$=0.962, $k_{-1}$=1.106, $k_2$=0.498, $k_{-2}$=0.000452, which gave a fit score of 0.0011. Model parameters for LiCre (B,D) were $\alpha$=1.71, $RUmax_h$=53.2, $RUmax_f$=38.1, $k_f$=0.099, $k_1$=0.98, $k_{-1}$=0.481, $k_2$=4.655, $k_{-2}$=0.00286, which gave a fit score of 0.00193. This experiment was performed once.

fast, while the dissociation was slightly faster (Fig. 2B). This demonstrates that LiCre is able to bind its DNA target site even if it is not photoactivated, which may explain why the blue-light treatment did not markedly increase the detection of LiCre chromatin-binding *in vivo* (Fig. 1).

The profiles obtained with half *loxP* sites were also very informative. In accordance with Rüfer et al. (2002), Cre dissociation from the half site was much faster than its dissociation from the full site (Fig. 2C). Similarly, for LiCre, dissociation from the half site is also faster, but the difference is not as pronounced as for Cre (Fig. 2D). This suggests cooperativity between the two Cre or LiCre units when binding to the full *loxP* site and that cooperativity of LiCre in the dark may be lower than that for Cre.

### LiCre binding to DNA is strong and cooperative

To rationalize our *in vitro* measurements in terms of binding affinities, we developed a kinetic model of the association and dissociation of LiCre and Cre molecules to DNA, accounting for the SPR protocol (see Materials and Methods). Briefly, based on Rüfer et al. (2002), the binding of one molecule to a half site is described as a simple one-step assembly reaction, while the binding of two molecules to a full site is assumed to happen sequentially: the binding of the first molecule is

assumed to follow the same kinetics as for a half site, while the second binding event may have different kinetics due to cooperativity mediated by the presence of the already-bound molecule (Fig. 3A). In addition to the association ($k_1$, $k_2$) and dissociation ($k_{-1}$, $k_{-2}$) rates of each step, the model comprises the following parameters that enable its direct confrontation to experimental acquisitions: a coefficient of mass transport in the biosensor chip ($k_t$), a linear coefficient ($\alpha$) linking the response unit (RU) of the SPR experiments to the amount of bound proteins, and maximal RU values ($RUmax_h$ and $RUmax_f$) when all of the half or full *loxP* sites are fully occupied. We fitted these parameters by least-squares minimization between predicted and experimental sensorgrams (Fig. 2A-D, see Materials and Methods). To study the identifiability of model parameters, we ran the procedure 1000 times, starting from different initial parameter values picked randomly within acceptable ranges. Among the resulting 1000 trials, 541 and 859 led to high-quality fits for Cre and LiCre, respectively (Fig. 3B,C). The corresponding sets of parameters are shown in Fig. 3D-F. Fitted values of $RUmax_h$ and $RUmax_f$ had very low standard deviations and were therefore very well defined for both Cre and LiCre (Fig. 3F). Parameters $k_t$ and $\alpha$ were each well estimated for LiCre (Fig. 3E). For Cre, their individual estimation was less precise, but as anticipated by their definition, they covaried and their product

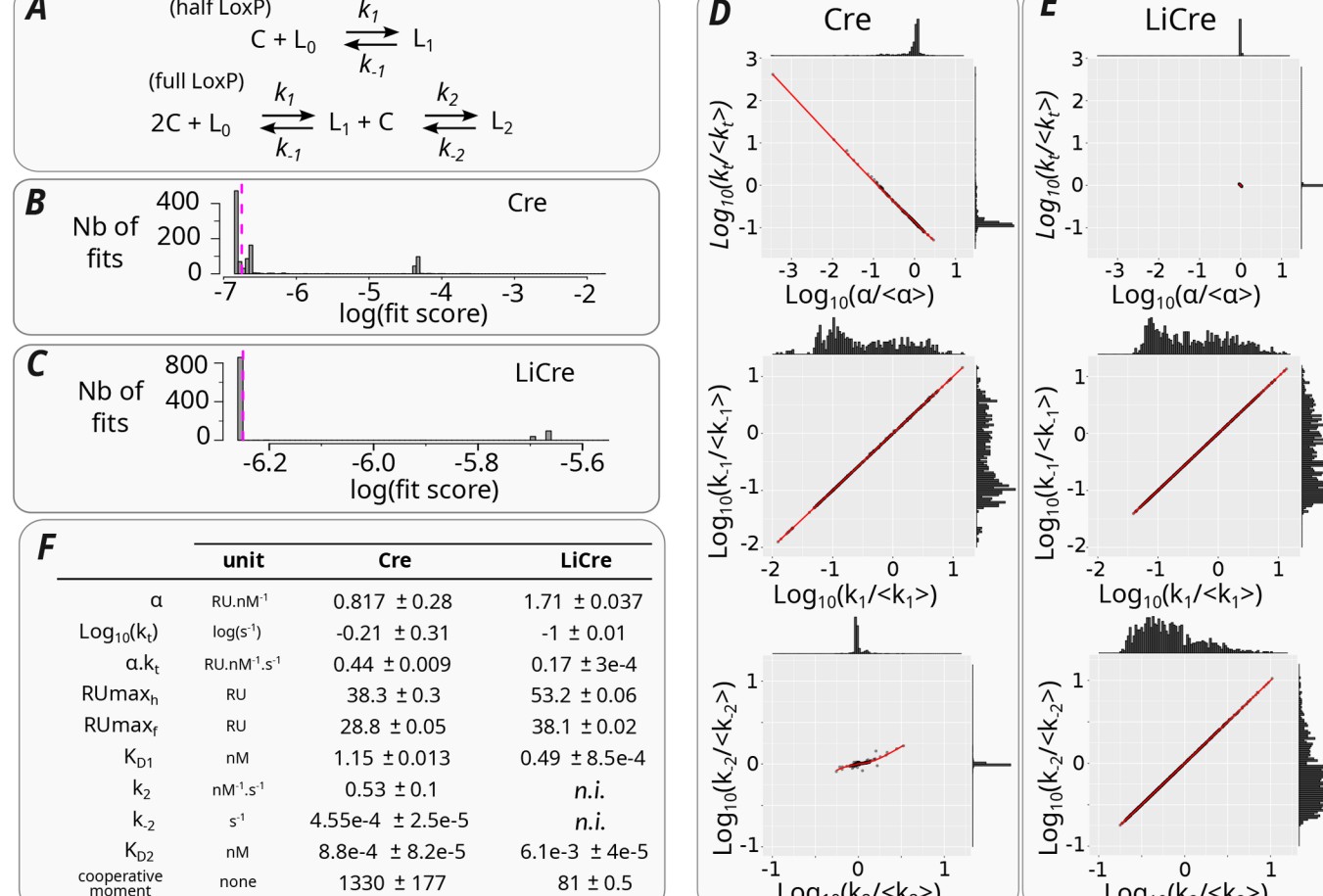

**Fig. 3. Kinetic modeling of DNA binding.** (A) Model for association of Cre or LiCre molecules (*C*) to DNA *loxP* site (*L*). $L_0$, $L_1$ and $L_2$ denote DNA sites occupied by zero, one and two molecules, respectively. (B,C) Distribution of fit scores across 1000 optimizations of the DNA-binding model for Cre (B) and LiCre (C). The lower the score, the better the fit (see Materials and Methods). Dashed magenta line, arbitrary threshold used to select high-quality fits. (D) Best estimates of model parameters obtained from 541 optimizations for Cre:DNA binding (see Materials and Methods). Every dot corresponds to one optimization. Histograms show marginal distributions of *x* and *y* values. Red line, LOWESS local regression. (E) Same representation as in D, showing results from 859 optimizations for LiCre:DNA binding. (F) Table of estimated parameter values (mean±s.d.). n.i., non-identifiable.

Table from panel F:

| | unit | Cre | LiCre |
|---|---|---|---|
| $\alpha$ | RU.nM$^{-1}$ | 0.817 ± 0.28 | 1.71 ± 0.037 |
| $Log_{10}(k_t)$ | log(s$^{-1}$) | -0.21 ± 0.31 | -1 ± 0.01 |
| $\alpha.k_t$ | RU.nM$^{-1}$.s$^{-1}$ | 0.44 ± 0.009 | 0.17 ± 3e-4 |
| $RUmax_h$ | RU | 38.3 ± 0.3 | 53.2 ± 0.06 |
| $RUmax_f$ | RU | 28.8 ± 0.05 | 38.1 ± 0.02 |
| $K_{D1}$ | nM | 1.15 ± 0.013 | 0.49 ± 8.5e-4 |
| $k_2$ | nM$^{-1}$.s$^{-1}$ | 0.53 ± 0.1 | *n.i.* |
| $k_{-2}$ | s$^{-1}$ | 4.55e-4 ± 2.5e-5 | *n.i.* |
| $K_{D2}$ | nM | 8.8e-4 ± 8.2e-5 | 6.1e-3 ± 4e-5 |
| cooperative moment | none | 1330 ± 177 | 81 ± 0.5 |

was robustly estimated (Fig. 3D,F). Similarly, association and dissociation rates of the first binding reaction were not individually identifiable, but their ratio ($K_{D1}=k_{-1}/k_1$, *aka* dissociation constant) was (Fig. 3D-F). The reaction rates of the second binding reaction could be identified for Cre but not for LiCre; however, their ratio ($K_{D2}=k_{-2}/k_2$) was precisely estimated for both proteins (Fig. 3D-F). For Cre, these dissociation constants ($K_{D1}=1.15$ nM and $K_{D2}=8.8\times10^{-4}$ nM) highlight a very strong cooperativity ($K_{D2}\ll K_{D1}$) and are consistent with those previously reported (Rüfer et al., 2002; Ringrose et al., 1998) (see Discussion). Importantly, the cooperativity for LiCre is also strong ($K_{D1}=0.49$ nM and $K_{D2}=6.1\times10^{-3}$ nM, cooperative moment $K_{D1}/K_{D2}=81$) but much weaker than for Cre ($K_{D1}/K_{D2}=1330$). This is consistent with the way LiCre was derived from Cre, which involved the destabilization of protein:protein binding, both by introducing point mutations in the αN helix required to lock C-terminal domains together and by truncating and hindering the αA helix engaged in the interaction between N-terminal domains (Duplus-Bottin et al., 2021) (see Discussion).

## Experimental quantification of LiCre-*loxP* recombination following various regimes of photo-stimulation

To quantify recombination in live cells, we used a yeast strain harboring a previously described quantitative reporter (Duplus-Bottin et al., 2021). This system is based on the LiCre-mediated excision of a transcriptional terminator preventing the expression of a fluorescent protein (Fig. 4A). This excision occurs by recombination between *loxP* sites located 1.3 kb from one another. In a typical experiment, a population of yeast cells is illuminated with chosen doses and dynamics of blue light; cells are then resuspended in fresh medium in the dark to let them express the fluorescent protein if recombination occurred; the population is finally analyzed by flow cytometry to determine the fraction of fluorescent cells. This fraction corresponds to the proportion of stimulated cells that achieved recombination, therefore providing a clear estimate of recombination efficiency. To control for any potential bias associated with the chromosomal locus where recombination takes place, every cell carried two versions of the reporter: one on chromosome IV encoding mCherry (red fluorescence) and one on chromosome II encoding GFP (green fluorescence). LiCre expression was conferred by a plasmid introduced into the resulting strain.

We first evaluated the phototoxicity of pulsed blue light on yeast viability by applying illumination to a suspension of cells and by counting colony-forming units (Fig. 4B). This showed that yeast viability is not affected by our illumination conditions.

We then verified the ability of the two fluorescent reporters to distinguish negative from positive cells by flow cytometry. As shown in Fig. S3, although both fluorescent proteins were reliably detected, the autofluorescence of negative cells was lower for mCherry, conferring a higher signal/noise ratio than the GFP channel. We therefore used, in most of our assays, the frequency of red cells as the estimate of recombination efficiency.

We applied this assay to compare the efficiency of LiCre with its epitope-tagged version LiCre-V5. After 1 h of photo-stimulation (1-s pulses of light applied every 10 s), the fraction of cells that had recombined the mCherry reporter (ON cells) was ~40% in the case of LiCre and only ~20% in the case of LiCre-V5 (Fig. 4C). This difference in efficiency was statistically significant (*P*-value=0.013, Welch *t*-test); it showed that the presence of repeated V5 epitopes at the C-terminal end of LiCre reduces its activity. It is possible that this tag negatively affects the DNA-binding affinity of LiCre, which would explain the relatively low ChIP signal observed above for

LiCre-V5 *in vivo* (Fig. 1) as compared to the strong DNA-binding affinity measured *in vitro* for LiCre (Figs 2,3).

We then evaluated whether the photoactivation of LiCre could be limited by the availability of intracellular flavin mononucleotide (FMN). FMN is an essential cofactor for the photoinduced conformational switch of LiCre's LOV domain. Cells produce it by phosphorylation of its precursor riboflavin. To test if LiCre efficiency would increase in conditions that maximize FMN production, we applied our assay on cells cultured in the presence of various concentrations of riboflavin (up to 20 µM). As shown in Fig. 4D, the presence of extracellular riboflavin had no significant effect on recombination efficiency. To exclude the possibility that riboflavin was insufficiently processed into FMN, we added in the LiCre-encoding plasmid a cassette encoding the overexpression of the yeast riboflavin kinase Fmn1p (constitutive expression conferred by the Padh1 promoter). In this case, a modest increase in LiCre efficiency was seen at 2 and 5 µM riboflavin, but not at higher concentrations (10 and 20 µM) (Fig. 4D). We conclude that, in standard conditions, FMN is probably sufficiently abundant in yeast cells for the optimal activation of LiCre.

Fig. 4E shows how recombination efficiency increased with the duration of a defined pulsed stimulation. In agreement with previous observations (Duplus-Bottin et al., 2021), 10 min was enough to induce recombination in a significant proportion (~10%) of cells. In general, we counted slightly more red cells than green cells. This difference could indicate that the reporter locus on chromosome IV is more favorable to recombination than its counterpart on chromosome II. To test for this possibility, we swapped the location of the two reporters – placing the GFP reporter on chromosome IV and the mCherry reporter on chromosome II – and we compared recombination efficiencies between the resulting strain and the strain with the original locations. Each reporter displayed the same efficiency at the two integration loci (Fig. 4F); this efficiency being consistently lower for the GFP reporter than for the mCherry reporter. The genomic locus is therefore not the determinant of the different reported efficiencies. Instead, and as supported by the higher signal/noise ratio of the red reporter (Fig. S3), we believe that this difference results from different measurement accuracies.

Importantly, we never observed recombination in 100% of the cells. After prolonged (>3 h) photo-stimulation, the proportion of positive cells reached a plateau of about 85%. Most of these cells displayed both red and green fluorescence, leaving a remaining ~15% cells negative for both reporters. This suggested that the upper limit of recombination efficiency was caused by a cell-specific factor rather than a locus-specific factor. We hypothesized that this upper limit could, at least in part, result from the occasional loss of the LiCre-encoding plasmid.

To test this possibility, we repeated the experiment, but instead of illuminating the population of cells, we plated diluted samples of it on both selective and nonselective solid media (Table S6). This revealed that, in our experimental conditions, the proportion of cells having lost the LiCre plasmid at the time of illumination was between 10% and 30%. Thus, we cannot expect the proportion of cells undergoing LiCre-mediated recombination to exceed 70%-90% in our assay, even after prolonged illumination.

We also observed substantial variability between experiments. For example, a pulsed stimulation of 60 min generated nearly 50% of positive cells in the experiment of Fig. 4E, whereas the same stimulation generated only ~30% of positive cells in another experiment (see figure 3G of Duplus-Bottin et al., 2021). We suspected that variations in room temperature could contribute to this variability and we tested this possibility by quantifying

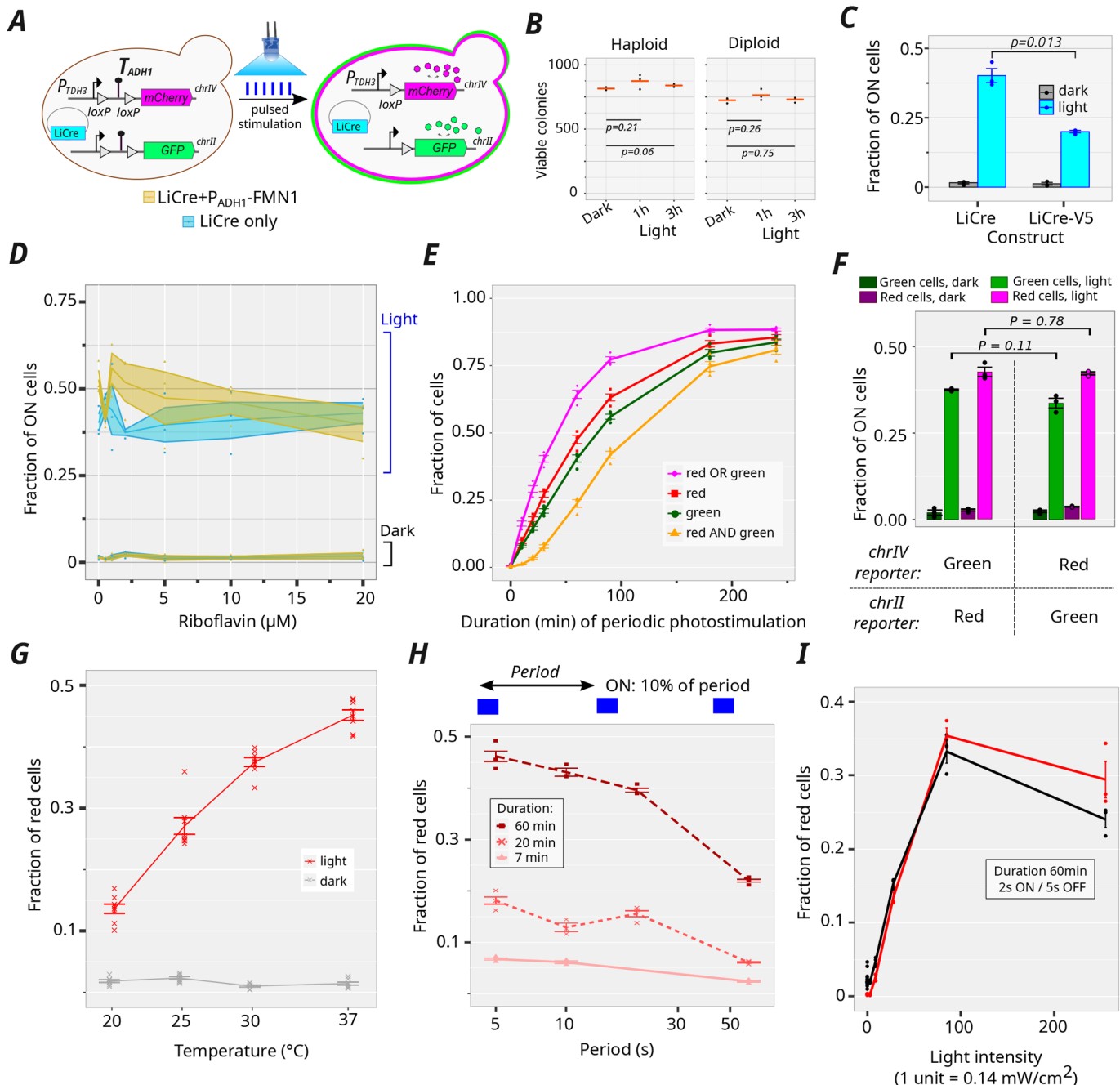

**Fig. 4.** See next page for legend.

recombination efficiency in various thermo-controlled conditions. The same pulsed stimulation generated between ~15% and ~45% positive cells when temperature varied between 20°C and 37°C, respectively (Fig. 4G), with higher temperatures leading to more efficient recombination, thus explaining most of the variability mentioned above. We therefore conducted subsequent experiments under strict temperature-controlled conditions (30°C), which ensured reproducibility across independent datasets (Fig. S4).

To further explore the dynamic properties of the LiCre-*loxP* reaction *in vivo*, we applied pulsed photo-stimulation at various periods with two different duty cycles (fraction of the period when light is on). The list of experiments is summarized in Table S5 and only a subset of the data is shown in Fig. 4H. For a fixed total cumulative time of light exposure, recombination efficiency

declined when the period increased. For example, applying a total illumination of 6 min (dark red in Fig. 4H) led to an efficiency above 40% when the time between consecutive pulses of light was short (4.5 s) but below 25% when this period was longer (54 s). This may reflect the existence of a lag time, in the order of tens of seconds, before LiCre returns to its dark, inactive state in the absence of light.

In order to quantify the photosensitivity of the reaction, we illuminated yeast cells at various light intensities. We chose a regime where light pulses were long enough to ensure efficient activation (2 s) and frequent enough to prevent the system from reverting to its inactive state (every 7 s). As expected, we observed an overall increase in recombination efficiency with light intensity from 0 to 12 mW/cm$^2$ (Fig. 4I, red). Intriguingly, the fraction of positive cells was slightly lower when we applied maximal intensity

**Fig. 4. Experimental quantification of *in vivo* LiCre-*loxP* recombination following various photo-stimulation conditions.** (A) Reporter system. Yeast cells harbor two reporter constructs integrated in their genome and express LiCre from a plasmid. A transcriptional terminator ($T_{ADH1}$), flanked by *loxP* sites, prevents expression of mCherry or GFP fluorescent proteins. LiCre photo-stimulation triggers recombination, which removes $T_{ADH1}$ and releases expression. The rate of recombination events can be estimated by counting fluorescent cells in a flow cytometer. (B) Viability assay of phototoxicity. Strain GY1761 (haploid) or GY2517 (diploid) was cultivated for 18 h in three independent cultures (biological replicates); the saturated cultures were transferred to polystyrene flat-bottom well plates and illuminated or not (dark control) during 1 or 3 h by pulses of blue light (450 nm, period of 10 s, duty cycle of 10%, intensity of 35 mW/cm²). For each culture, cell dilutions were plated on three plates to count colony-forming units (viable colonies). Every dot represents the mean number of colonies on the three plates for one biological replicate. Red bar: mean. *P*-value: two-tailed Welch *t*-test on biological replicates. This experiment was performed once. (C) Recombination rate induced by LiCre and LiCre-V5 following pulsed photo-stimulation. Cells were subjected to pulses of blue light (450 nm, period of 10 s, duty cycle of 10%, intensity of 35 mW/cm²) for 1 h and then transferred to fresh medium for 4 h to enable expression of the fluorescent proteins. The fraction of cells expressing the mCherry reporter (ON cells) was determined by flow cytometry. *P*-value: two-tailed Welch *t*-test, *n*=3 biological replicates per condition. (D) Recombination rate induced by LiCre in conditions of riboflavin supplementation. Same illumination conditions as in C. Medium was supplemented with various concentrations of riboflavin (*x*-axis). Blue, LiCre expressed under the Pmet17 promoter (plasmid pGY466). Red, LiCre expressed under the Pmet17 promoter and FMN1 expressed under the Padh1 promoter (plasmid pGY781). Dots: independent cultures (biological replicates). Lines and ribbons: mean±s.e.m., *n*=3 biological replicates per condition. (E) Recombination rate following pulsed stimulation of various duration. Cells (strain GY2214) were subjected to pulses of blue light (450 nm, period of 10 s, duty cycle of 5%, intensity of 35 mW/cm²) for the indicated amount of time and then transferred to fresh medium for 4 h to enable expression of the fluorescent proteins. Green, red and nonfluorescent cells were then counted by flow cytometry; *n*=3 biological replicates. (F) Comparison of recombination efficiencies estimated at the two integration loci. Strains GY2214 and GY2517 were compared. They harbor the same two reporters but at positions that were swapped between two genomic loci. Cells were photo-stimulated as in C. *P*-value: two-tailed Welch *t*-test on *n*=3 biological replicates. (G) Same as in E but for a single duration and various temperatures (one dot per biological replicate). (H) Same as in E but with various periods and a duty cycle of 10%. Intensity of 35 mW/cm², *n*=3 biological replicates. (I) Same as in E and *h* but with various light intensities, a period of 7 s, a duty cycle of 29% applied for 60 min, *n*=3 biological replicates. Colors correspond to two independent sets of experiments. In *e-i*, replicates (dots) correspond to independent cultures, each inoculated with an independent transformant of the LiCre plasmid. Lines connect mean values. Error bars: mean±s.e.m. Every experiment shown in C-I was performed once, at the dates indicated in Table S5.

(35.6 mW/cm²). This observation was reproduced when we repeated the experiment (Fig. 4I, black, Dataset 11 listed in Table S5), possibly pointing to unknown inhibitory effects under prolonged or intense illumination. It is possible that high-intensity blue light causes damage to LiCre itself. It is also possible that, even if it does not compromise cell viability (Fig. 4B), intense illumination triggers a cell stress response that inhibits recombination efficiency.

Overall, the experimental data generated here describe the efficiency of LiCre following illumination conditions varying in duration, in dynamics or in light intensity. This dataset associates the kinetic properties of the input stimulus with the quantitative output of the system.

## Kinetic modeling of LiCre-*loxP* recombination
To describe the light-induced recombination catalyzed by LiCre, we built a mathematical model based on three steps (Fig. 5, see Materials and Methods).

First, the association of LiCre to DNA occurs in the dark and is at equilibrium. Using the dissociation constants inferred from the SPR experiments (see above), we plotted the probabilities of occupancy of the four LiCre binding sites located on two *loxP* sequences, assuming independent binding at the two *loxP* loci (Fig. 5). At LiCre concentrations above 1 nM, over 97% of cells have both their *loxP* sites fully occupied (four bound LiCre units in total).

Second, during stimulation by light, bound LiCre units are activated at a rate $k_{ON}$. In the dark condition, they deactivate at a rate $k_{OFF}$. Given these rates and the dynamics of illumination, we can compute the probability $P_{act}(t)$ that a given LiCre unit is active at time $t$ (Fig. 5C, top). Since LiCre is activated by the asLOV2 switch, we chose $k_{ON}$ as previously reported for the LOV2 domain (Renicke et al., 2013). This rate increases linearly with the incoming flux of photons and is in the order of per second (see Materials and Methods).

Third, if the two *loxP* sites are both fully occupied and if at least $x$ among the four LiCre units are active, with $x$ ranging from 1 to 4, recombination can occur at a constant rate $R_0$ that encompasses the loop probability between the sites, the synapse formation and the actual catalytic reaction of recombination. All this leads to an overall time-dependent recombination rate $R(t)$ that is proportional to $R_0$, to the probability of having four LiCre bound to the *loxP* sites and to the probability of having at least $x$ light-activated LiCre molecules, which is a function of $P_{act}(t)$ (Fig. 5A).

The well-known Cre-*loxP* reaction is reversible, and this is presumably also the case for the LiCre-*loxP* reaction. As described above, we concluded from our experimental data that all cells having kept the LiCre plasmid would eventually achieve recombination after prolonged photoactivation. Thus, at equilibrium, the backward reaction (reinsertion of the excised circular DNA) is unlikely and outcompeted by the forward reaction, and we therefore neglected it in our model.

A reaction simulated with this model is shown in Fig. 5C. In this example, periodic pulses of light were applied, and the fraction of activated LiCre units reached a periodic steady state after only 60 s, oscillating around ~80% (Fig. 5C, top) with the light stimulus. The overall time-dependent reaction rate $R(t)$ also fluctuates, the speed of recombination being highly affected by the number ($x$) of active LiCre units needed to form a functional complex (Fig. 5C, center). After plotting the cumulative fraction of recombined molecules over time (Fig. 5C, bottom), we concluded that this three-step model provides realistic outcomes that can be compared to experimental data.

## Model fitting to experimental data suggests that photoactivation of two LiCre protomers may be sufficient to trigger recombination
We investigated whether the above model could explain the efficiencies of DNA recombination that we observed in yeast cells stimulated under various conditions. In this yeast assay, the fraction of cells that switched ON fluorescence directly reports on the fraction of DNA molecules (among the total cell population) that underwent recombination. The corresponding experimental data can therefore be compared with model predictions.

Although we did not quantify LiCre concentration in yeast cells, we estimated from the literature (see Materials and Methods) that it was probably above ~1 nM, conferring near-complete occupancy at the two *loxP* sites *in vitro* (Fig. 5B). Arbitrarily, we decided to set LiCre concentration in the model at 1 nM. Note that this choice does not impact the quality of the fit nor the order of magnitudes of the inferred parameters (see Materials and Methods).

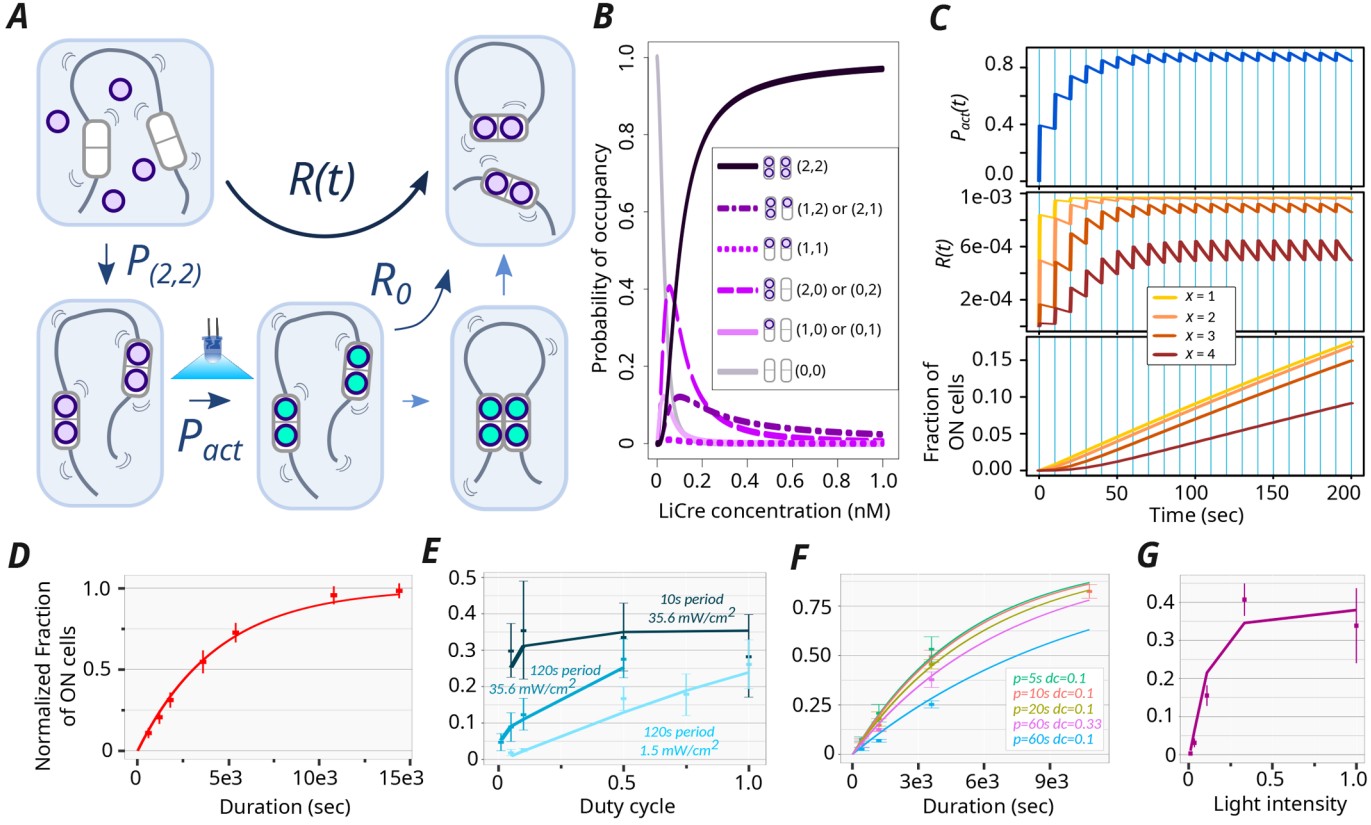

**Fig. 5. Kinetic model of LiCre-*loxP* recombination.** (A) Model description. Purple and cyan beads represent inactive and active LiCre units, respectively. Occurrence of the reaction assumes the binding of four LiCre units [which happens with probability $P_{(2,2)}$], the photoactivation of at least $x$ of them, with $x \in$ {1,2,3,4}, and a maximal reaction rate $R_0$ once the complex can form. Reaction rate $R(t)$ summarizes the triggering of the reaction by light. See text and Materials and Methods for details. (B) Effect of [LiCre] concentration on the probability of occupancy at two *loxP* sites. $(n,m)$ indicates $n$ and $m$ LiCre molecules bound to the first and second *loxP* sites, respectively. (C) Simulations under periodic pulses of light. Light was ON during the vertical blue stripes; each pulse occurred every 10 s and lasted 0.5 s. [LiCre]=1 nM, $k_{ON}$=1 s$^{-1}$, $k_{OFF}$=0.007 s$^{-1}$, $R_0$=10$^{-3}$ s$^{-1}$. From top to bottom: probability that one LiCre unit is active ($P_{act}$); reaction rate at each possible value of $x$ (required number of activated units); cumulative fraction of recombined molecules over time at each possible value of $x$. (D-G) Best model prediction using $x$=2 and parameters simultaneously fitted to all experimental data. Lines, model simulations. Dots and bars, experimental data (mean±2 standard deviations), $n{\geq}3$ biological replicates. Each panel corresponds to one consistent series of acquisitions, where the fraction of mCherry-positive yeast cells that underwent recombination was measured by flow cytometry (Datasets 1-10 listed in Table S5). Value $k_{OFF}$ inferred from all data: 50 s$^{-1}$. (D) Effect of the duration of illumination (same data as in Fig. 4E). Inferred $R_0$ value: 3.5.10$^{-4}$ s$^{-1}$. (E) Effect of illumination dynamics and intensity (duration of 1 h), data from figure 3F-G of Duplus-Bottin et al. (2021). Inferred $R_0$ value: 0.000125 s$^{-1}$. (F) Effect of illumination dynamics and duration. Inferred $R_0$ value: 0.00014 s$^{-1}$. (G) Effect of illumination intensity (same data as in Fig. 4H, red). One unit=35.6 mW/cm$^2$. Inferred $R_0$ value: 0.00022 s$^{-1}$.

In total, three parameters of the model are unknown and can be optimized to fit experimental data: $k_{OFF}$, the speed of LiCre inactivation in the dark; $x$, the minimal number of active LiCre protomers in the complex required for the recombination reaction to occur; and $R_0$, the reaction rate of the enabled synapse. We ran the optimization of $k_{OFF}$ and $R_0$ for each value of $x$ among {1, 2, 3, 4} over a rich set of experiments performed under various conditions of illumination (different durations, periods, duty cycles and light intensities, Datasets 1-10 listed in Table S5). Given the batch-to-batch experimental variability mentioned above, we optimized

$R_0$ specifically for each series of experiments. Finally, to account for the frequent loss of the LiCre plasmid (see above), we normalized experimental data so that the highest observed efficiency was 100% (see Materials and Methods).

Results from parameter optimization are summarized in Table 1. Inferred $R_0$ values ranged from 1.25 to 4.10$^{-4}$ s$^{-1}$ (characteristic time from 42 min to 2.2 h) independently of $x$, and this variation primarily corrected for the abovementioned suspected effect of temperature in one of the datasets. When $x$ was set to either 1, 2, 3 or 4, the inferred $k_{OFF}$ rates corresponded to characteristic LiCre

**Table 1. Best-fit parameters of the recombination reaction model**

| Parameter | Dataset specificity | $x$=1 | $x$=2 | $x$=3 | $x$=4 |
|---|---|---|---|---|---|
| $k_{OFF}$ (10$^{-3}$ s$^{-1}$) | All | 125.6 | 50 | 19.9 | 7.1 |
| $R_0$ (10$^{-4}$ s$^{-1}$) | Duration, Dataset 7 (Fig. 5D) | 3.5 | 3.5 | 3.5 | 4.0 |
| | Dynamics, Datasets 1 and 2 (Fig. 5E) | 1.25 | 1.25 | 1.25 | 1.4 |
| | Dynamics, Datasets 3, 4, 5 and 6 (Fig. 5F) | 1.25 | 1.4 | 1.4 | 1.6 |
| | Intensity (Fig. 5G) | 2.2 | 2.2 | 2.0 | 2.2 |
| Fit score* | All | 0.034 | 0.030 | 0.033 | 0.041 |

*Chi-square-like score quantifying distance to observations. The lower this score, the better the fit (see Materials and Methods).

deactivation times of about 8 s ($k_{OFF}=125.6\times10^{-3}$ s$^{-1}$), 20 s ($k_{OFF}=50\times10^{-3}$ s$^{-1}$), 50 s ($k_{OFF}=19.9\times10^{-3}$ s$^{-1}$) and 140 s ($k_{OFF}=7.1\times10^{-3}$ s$^{-1}$), respectively. It has long been known that, once photoactivated, the LOV2 domain needs tens of seconds to return to its dark state (Swartz et al., 2001). It is therefore unlikely that LiCre can be deactivated after only 8 s in the dark as predicted by the model where a functional recombination complex is formed if only one LiCre unit is active ($x=1$). Notably, adequacy of model predictions to the data was maximized for $x=2$ (Table 1), with an overall very good fit (Fig. 5D-G), and using $x=4$ provided the least adequate match to experiments but still captured well most of the data (Table 1). This suggests that the complex may not need four active LiCre units to be functional and that photoactivation of only two units may be sufficient.

### The model captures the effect of a point mutation affecting the light cycle of LiCre

We tested if the response of LiCre to pulsed stimulation was modified by mutations altering the light cycle of its asLOV2 domain. Several point mutations were previously described that either slow down or accelerate the recovery of asLOV2 in the dark (Nash et al., 2008; Christie et al., 2007; Song et al., 2011; Kawano et al., 2013). Using earlier reports, we selected three mutants with faster recovery (V416T, N425Q/I427V and T418S) (Kawano et al., 2013) and three mutants with slower recovery (V416L, F494I/L496S, and Q513L) (Nash et al., 2008; Kawano et al., 2013). We introduced these mutations in LiCre, and using our yeast assay, we quantified the recombination efficiency of the resulting six variants under a given regime of periodic photo-stimulation. All mutations severely affected the performance of LiCre (Fig. 6A). Photoactivation was lost in the slow-recovery mutant Q513L. Four mutants (V416L, F494I/L496S, V416T and N425Q/I427V) showed a pronounced residual activity in the dark. Photoactivation was preserved in the remaining mutant, T418S, but it was reduced as compared to wild-type (WT) LiCre.

We reasoned that if this reduction resulted from a faster recovery of LiCre to its inactive state, then increasing the duration or the frequency of the light pulses would restore activity in this mutant,

possibly up to WT levels. To test this prediction, we quantified the recombination efficiencies of both LiCre$^{WT}$ and LiCre$^{T418S}$ following photo-stimulation regimes for two different periods and three different duty cycles (Fig. 6B). As expected, for a fixed duty cycle, the recombination efficiency of both constructs was higher under a short (10 s) period than under a long (60 s) period; and similarly, for a fixed period, it increased with the duty cycle, especially for the 60-s period. The main difference between the two variants was seen at 10% duty cycle and a period of 60 s. Under this condition, the mean fraction of switched cells was 0.355 for WT LiCre and 0.246 for the T418S mutant, a difference that was statistically significant (P-value=0.005, Welch t-test). At 50% duty cycle and for both periods, the two constructs displayed similar efficiencies. Thus, the effect of the T418S mutation manifests only in sub-optimal regimes where the time for inactivation in the dark is long. This observation is consistent with the prediction that the T418S mutation increases the recovery rate of LiCre.

Given this consistency, we evaluated whether our kinetic model was able to capture such a difference in recovery rate between the two constructs. By fitting the comparative experiments described above (Dataset 13 listed in Table S5) with our kinetic model, we inferred $k_{OFF}$ and $R_0$ parameters for LiCre$^{WT}$ and LiCre$^{T418S}$ independently. We did so separately for all four $x$ values (=1, 2, 3 or 4, the required number of active units). Corresponding model predictions for $x=2$ are shown in Fig. 6B, but all parameter estimates are provided in Table S7. All four models inferred $k_{OFF}$ values that were higher for the mutant than for the WT. This difference increased from 1.58-fold to 1.99-fold when $x$ increased from 2 to 4. This trend is consistent with the fact that a model imposing that all four units must be active for the reaction ($x=4$) is more sensitive to the inactivation rate ($k_{OFF}$) of individual units. From the different estimates of $k_{OFF}$ between the LiCre variants, we conclude that the faster asLOV2 recovery caused by the T418S mutation translates into a ~1.6-fold faster inactivation of LiCre in the dark.

In conclusion, our three-step model can recapitulate the recombination rates observed under various dynamics and intensities of light stimulation; it suggests that synapse assembly may occur even if not all four LiCre units are photoactivated; it

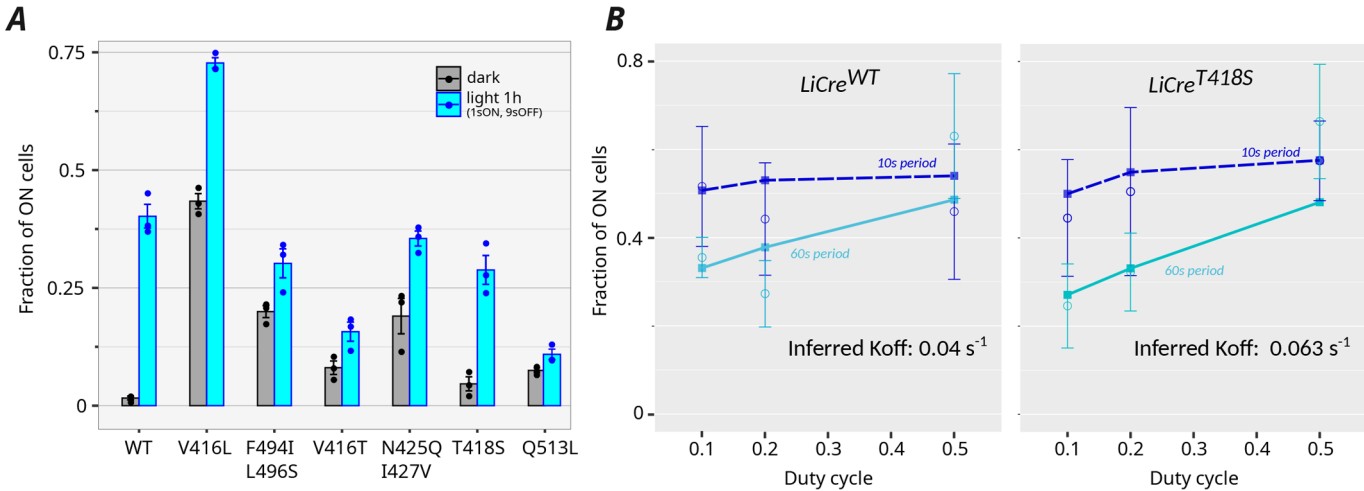

**Fig. 6. Performance of LiCre mutants with altered photo-cycle of the LOV domain.** (A) Recombination induced by wild-type (WT) LiCre and six genetic LiCre variants. Efficiency (fraction of ON cells) was quantified by flow cytometry following pulsed photo-stimulation (450 nm, period of 10 s, duty cycle of 10%, intensity of 35 mW/cm$^2$, duration of 1 h); n=3 biological replicates. (B) Recombination efficiencies of LiCre WT and T418S variant following various regimes of pulsed photo-stimulation (Dataset 13 of Table S5). Circle dots: mean values of experimental data (n=5 biological replicates) for a duration of 60 min. Error bars: ±standard deviation of experimental data. Squares connected by lines: predictions from model with x=2 and best-fit parameters $R_0$ and $k_{OFF}$ (Table S7).

provides quantitative estimates of the reaction rate of the assembled synapse and of the speed of the monomer's recovery to the dark state; and it captures an expected faster recovery resulting from the T418S point mutation of the LOV domain of LiCre.

## DISCUSSION

By coupling modeling with experiments conducted both *in vitro* and *in vivo*, this work provides knowledge on the kinetics of site-specific DNA recombination by blue-light photo-stimulation of the LiCre-*loxP* system. We found that LiCre has high affinity for its target *loxP* substrate even if it is not photoactivated, that this binding is cooperative – although less than for Cre – and that the observed data are best explained by a model where the photoactivation of at least two LiCre protomers enables recombination. We quantified LiCre DNA-binding affinity, and we estimated characteristic times involved in the photoinduced reaction. We identified a point mutation altering the light cycle of LiCre.

Our *in vivo* and *in vitro* analyses of LiCre:DNA binding provided contrasting results. *In vivo*, LiCre binding to chromatin was detected by a weak (but reproducible) ChIP signal, whereas *in vitro*, affinity of LiCre for its DNA target was very high. This contrast might result from several factors. First, naked DNA is not chromatin. It is possible that, *in vivo*, the presence of nucleosomes or other DNA-binding factors may penalize the accessibility of LiCre to the *loxP* site. Second, the buffer we used in our *in vitro* assay may not adequately reflect the microenvironment of the *loxP* locus in the yeast nucleus. Third, ChIP is an indirect method. It relies on a fixation step, on the properties of the anti-V5 antibody and on the proper accessibility of the V5 epitopes in the fragmented particles of cross-linked material; its quantification by qPCR provides estimations that are not absolute but relative (ChIP versus input, target locus versus reference loci). It is therefore very difficult to compare quantitatively the results obtained *in vitro* and *in vivo*. In particular, although LiCre:DNA binding was weaker than Cre:DNA binding both *in vivo* (lower ChIP signal) and *in vitro* (weaker cooperativity), it is difficult to know if the extent of this difference is the same in the two contexts.

The fact that LiCre binds its DNA target already before illumination probably explains why it is faster and more efficient than other photoactivatable Cre-*loxP* systems (Duplus-Bottin et al., 2021). These other systems were obtained by splitting Cre in two halves, which were fused to optogenetic dimerizers enabling their assembly in response to light (Hochrein et al., 2018; Kawano et al., 2016; Kuwasaki et al., 2022; Taslimi et al., 2016). Given that Cre binds DNA by capturing it between its N- and C-terminal domains (Van Duyne, 2015), it is unlikely that these split subunits can bind DNA before they are assembled together. In contrast, LiCre is made of a single peptide chain comprising both N- and C-terminal domains, which makes it compatible with DNA binding prior to illumination. In addition, the single-chain design of LiCre, together with its high affinity for its target DNA, predicts that it may enable efficient recombination over a wider range of intracellular concentrations than split-based designs. This advantage may be important when intracellular concentration of the enzyme is unknown or difficult to control.

Our SPR measurements allowed us to compare the DNA-binding affinities of LiCre and Cre. For Cre, we estimated $K_{D1}$ at 1.15 nM, which is concordant with previous reports where it was estimated at 1.7 nM by electrophoretic mobility shift assays (EMSA) (Ringrose et al., 1998) and at 1.8 nM by SPR (Rüfer et al., 2002). For $K_{D2}$, however, these previous studies reported values of 0.0135 nM for EMSA and 0.02 nM for SPR, which are much higher than the

0.001 nM found in the present work. This difference, which results in different estimates of DNA-binding cooperativity, can be explained by differences in $Mg^{2+}$ concentration between the different studies. Unlike previous reaction conditions that did not contain any $Mg^{2+}$ cations (Rüfer et al., 2002) or contained 2 mM $Mg^{2+}$ partially chelated by 1 mM EDTA (Ringrose et al., 1998), our running buffer contained 10 mM $MgCl_2$ and no EDTA. Magnesium as well as spermidine has long been known to increase the activity of Cre *in vitro* (Abremski and Hoess, 1984), and the previous SPR study reported a marked increase in cooperativity when one of these ingredients was added to the reaction. For example, the DNA-binding cooperativity moment of Cre increased from 86 to 1700 when 5 mM spermidine was added, and the authors reported that the effect of $Mg^{2+}$ was similar (Rüfer et al., 2002). Thus, the high cooperativity previously observed in the presence of $Mg^{2+}$ is comparable to the one we found here (cooperativity moment of 1300). We conclude that our measure of the DNA-binding affinity of Cre by SPR is consistent with previous estimates, which provides confidence in the novel measures obtained here on LiCre.

Similarly to Cre, LiCre showed very high affinity for *loxP*, and LiCre proteins bind cooperatively to DNA. However, when compared to Cre, we observed a twofold stronger binding of the first unit to DNA and a sevenfold weaker binding of the second unit, resulting in a much less pronounced cooperativity (Fig. 3F). Given that cooperativity results from physical interactions between protomers, this observation is consistent with the way LiCre was engineered. Indeed, the original protein–protein interactions known to take place between Cre protomers were destabilized in three ways (Duplus-Bottin et al., 2021). First, the αA helix of the N-terminal domain was truncated. Second, two point mutations destabilizing protein–protein interactions were introduced in the αN helix of the C-terminal domain. These point mutations likely affect the strength of the mechanism by which this helix was recently shown to participate in the DNA-binding cooperativity of Cre: in the absence of DNA, soluble Cre is monomeric and retains its own αN helix in *cis*, but in the presence of *loxP* DNA, the αN helix is released and becomes an anchor that stabilizes the association of the second unit (Unnikrishnan et al., 2020). This release of the αN helix probably also occurs for LiCre, but likely with a lower contribution to cooperativity because the two mutations make αN a weaker anchor. Finally, the remaining segment of the N-terminal αA helix of Cre was directly fused to the Jα helix of asLOV2. Thus, the LOV domain of LiCre is located in its N-terminal globule, away from the C-terminal αN helix, and the stabilization mediated by the C-terminal module is probably not light-dependent; in agreement with the remaining cooperativity of LiCre observed here in the dark. Photoactivation of LiCre is expected to result from conformational changes occurring at its N-terminal: in the dark, the chimeric Jα-αA helix is probably unable to interact with the partner protomer because it is presumably well folded and sequestered by the asLOV2 domain; illumination is expected to cause the release of this helix, allowing it to refold and participate in inter-unit interaction. Given the multiple changes that destabilize protein-protein interactions, it is not surprising to observe a weaker DNA-binding cooperativity for LiCre than for Cre. It would be interesting to measure the DNA-binding affinity of light-activated LiCre. We may expect an increase in cooperativity if the chimeric helix refolds in a conformation suitable for protein–protein interaction, but probably not to the extent of Cre given the C-terminal mutations. Such measurements require dedicated equipment that can simultaneously quantify molecular associations and illuminate samples in appropriate conditions, which was not possible on our SPR station.

Unlike classical genetics or drug-based treatments, optogenetics offers the possibility to induce an intracellular reaction dynamically with high temporal precision. In the present study, applying pulsed stimuli with various dynamics and feeding the resulting data into a simple kinetic model allowed us to derive kinetic information from a rather simple yeast-based assay. We could conclude that, following their photoactivation, LiCre units get deactivated in tens of seconds, that the LOV-targeting point mutation T418S accelerates this deactivation; that a functional LiCre-loxP synapse may be able to form if at least two of its four LiCre units are active and that following this activation, it achieves DNA recombination quite slowly with a characteristic time of tens to hundreds of minutes. It is interesting to compare these conclusions with the kinetic properties of Cre-mediated recombination, which were previously studied by tracking the motion of single tethered DNA molecules in vitro (Fan, 2012; Pinkney et al., 2012; Shoura et al., 2012). These studies showed that, with a constitutively active Cre protein, the limiting steps of the reaction were the assembly of a functional synapse and the catalytic activity of the complex. For example, for intramolecular recombination reactions between loxP sites separated by 3044 and 870 bp, transposing the estimates of reaction rates found in Shoura et al. (2012) to equivalents of the $R_0$ parameter of our model (see Materials and Methods) leads to values of $0.2\text{-}0.55\times10^{-4}$ s$^{-1}$, respectively, corresponding to characteristic times reaching several hundreds of minutes, much slower than our in vivo estimation for two loxP sites separated by 1300 bp. Since, for LiCre, the steps of synapse assembly and catalysis are not expected to be faster (LiCre was derived from Cre by reducing, rather than increasing, its propensity to assemble the complex and achieve the reaction), the main difference between in vitro (Shoura et al., 2012) and in vivo (this study) $R_0$ values likely lies in the probability of forming a DNA loop, bringing the two loxP sites together (a prerequisite to synapse formation). This probability depends on the mechanical properties of the inter-loxP DNA segment (Rosa et al., 2010; Shoura et al., 2020); it may thus differ strongly between a stiff inter-loxP segment of naked dsDNA in vitro and a softer, chromatinized one in vivo (Hajjoul et al., 2013; Ringrose et al., 1999). Nonetheless, we note that some of our conclusions remain to be validated by additional direct mechanistic investigations. Single-molecule analyses, such as those previously performed on Cre (Fan, 2012; Fan et al., 2015), would enable us to evaluate more complex models. For example, it is possible that the reaction rate of a fully activated synapse may be faster than the rate of a synapse containing only two or three active LiCre units. Also, in vitro kinetic data might better discriminate models with different $x$ values. Complex assembly after activation of two units is consistent with the need of two contact interfaces to bridge the two dimer-loxP complexes together; however, this prediction was supported here by relatively mild differences in fit scores to the yeast-based data (Table 1).

Remarkably, the model was able to capture a quantitative change in the inactivation rate ($k_{OFF}$) of LiCre resulting from a point mutation targeting its LOV domain (Fig. 6B). This mutation, referenced as T418S on the phototropin 1 sequence, was initially isolated from a screen where asLOV2 recovery was tracked by monitoring the fluorescence of bacterial colonies (Kawano et al., 2013). The recovery rate of this mutant was reported to be ~14 times faster than the recovery rate of WT asLOV2. Here, we find that introducing this mutation in LiCre increases its inactivation rate to about 1.6-fold. It is important to note that the rates of asLOV2 recovery and the rate of LiCre inactivation do not describe the exact same biochemical steps. The former corresponds to the conformational change of the LOV domain from a photo-stimulated (light) to its original (dark) state; the

latter corresponds to the loss of activity of the full LiCre protomer, which involves not only the conformational change of its LOV domain but also the resulting putative loss of functional interaction with other LiCre protomers of the recombination synapse. It is therefore not surprising to observe a milder effect of the mutation on the speed of LiCre deactivation than on the speed of asLOV2 recovery.

Finally, our results have practical implications for experimental protocols employing LiCre. First, given its high affinity for loxP (Fig. 5B), overexpressing LiCre at high levels will probably not increase its efficiency. On the contrary, depending on how cells regulate the production of the FMN, an overproduction of LiCre might generate a proportion of LiCre molecules lacking this cofactor essential for photoactivation. Here, in the conditions used in our yeast-based assay, FMN is probably abundant enough in the cell to occupy all LiCre molecules. Otherwise, adding riboflavin to the culture medium and overexpressing the riboflavin kinase would have generated a higher fraction of switched cells in the assay, which was not observed (Fig. 4D). In mammalian cells, FMN was previously quantified at about 1 amol per cultured cell (Hühner et al., 2015) (equivalent to 600,000 molecules, or 1 µM, assuming a cell volume of 1 pL). Thus, it may be judicious to express LiCre at moderate levels (<1 µM) to ensure stoichiometry with FMN while still maximizing its binding to loxP. However, it is also possible that, if high levels of LiCre would cause a sequestration of FMN from the cellular pool, then cells might be able to respond by increasing FMN synthesis. This would provide enough cofactor for most LiCre molecules regardless of the expression level of LiCre.

Second, LiCre was three times more efficient at 37°C than at 20°C, with no additional residual activity in the dark (Fig. 4F). This observation is concordant with previous in vitro measurements for Cre (Buchholz et al., 1996). We therefore invite LiCre users to conduct experiments at 37°C whenever possible.

Third, our data indicate how to minimize the amount of blue light shone on cells – and its potential damaging effects thereof – while keeping LiCre induction maximal. With a 450 nm wavelength, we recommend using a photon flux of 12 mW/cm² and periodic pulses of 1 s spaced every 10 s.

In conclusion, by precisely characterizing the kinetic properties of the optogenetic LiCre-loxP recombination, this study provides useful insights for conducting targeted genetic manipulations of live cells using light.

## MATERIALS AND METHODS
### Strains and plasmids

Plasmids, strains and oligonucleotides used in this study are listed in Tables S1-S3. Records of laboratory stocks were traced using MyLabStocks (Chuffart and Yvert, 2014). Plasmids used for the production of recombinant LiCre (pET-LiCre=pGY611) or Cre (pET-Cre=pGY709) are made available from Addgene (https://addgene.org) under accession IDs 219797 and 219798, respectively.

To perform ChIP, we tagged LiCre with V5 epitopes (IPNPLLGLD) (Dunn et al., 1999). We placed a Gly-Ser flexible linker at the C-terminus of LiCre in order to avoid interference between these epitopes and the C-terminal αN helix known to be critical for LiCre activity (Duplus-Bottin et al., 2021). Nine repeats of the V5 epitope were amplified, using primers 1Q46 and 1Q47, from a plasmid kindly provided by Pascal Bernard. The resulting amplicon was co-transformed into yeast with plasmid pGY466 previously linearized by NdeI. The plasmid (LiCre-V5=pGY605) resulting from subsequent homologous recombination was recovered from yeast colonies and amplified in E. coli. We constructed Cre-V5 similarly by co-transforming in yeast the same amplicon with pGY502 linearized at NdeI. Subsequent rescue from yeast provided plasmid pGY607. Verification

by Sanger sequencing revealed that pGY605 contained nine V5 repeats as expected and pGY607 (used in the experiment shown in Fig. S1) unexpectedly contained twelve V5 repeats. We therefore repeated this construction. Sanger sequencing validated the presence of nine V5 repeats on the Cre-V5 plasmid (pGY820) used in the experiment shown in Fig. 1D.

For recombinant protein production, the LiCre and Cre coding sequences were codon optimized for *E. coli* and the corresponding sequences were synthesized and cloned in pET-19 by GeneCust to produce plasmids pGY611 (pET-LiCre) and pGY709 (pET-Cre), respectively. An mCherry reporter targeting the *LYS2* locus was obtained by cloning a *Kpn*I-*Avr*II fragment from pGY472 into the *Kpn*I-*Avr*II sites of pGY537, producing plasmid pGY618. For integration of a single *loxP* site at the *LYS2* locus, we built plasmid pGY621 by digesting pGY618 with *Bam*HI and *Age*I and applying Klenow fill-in and religation. LiCre mutants targeting the asLOV2 domain were generated by GeneCust via mutagenesis of plasmid pGY466. Note that positions V416, F494, L496, N425, I427, T418 and Q513 referenced on phototropin 1 correspond to physical positions V25, F103, L105, N34, I36, T27, and Q122, respectively, on the amino acid sequence of LiCre. An expression cassette consisting of the promoter of the *Saccharomyces cerevisiae* ADH1 gene, the coding sequence of *S. cerevisiae* FMN1, and the TEF terminator sequence from *Ashbya gossypii* was synthesized by GeneCust who cloned it into the *Kpn*I site of pGY466 to produce pGY781.

Reporter yeast strains were either GY2214 (Duplus-Bottin et al., 2021), carrying an mCherry reporter at the *HO* locus and a GFP reporter at the *LYS2* locus; strain GY2517, carrying the same mCherry and GFP reporters but at the *LYS2* and *HO* loci, respectively; or strain GY2753, carrying only the mCherry reporter at the *HO* locus. To obtain GY2517, we transformed GY855 with pGY618 linearized by *Nru*I and selected Lys−, Ura+ clones (pop-in) and then 5-FoA-resistant clones that had become Lys−, Ura− (pop-out). This produced strain GY2450, which we crossed with GY1761 to obtain GY2517. To obtain GY2753, we first cured the ade2-1 mutation of strain OAy470 (Aparicio et al., 1997) by transforming it with a PCR amplicon containing the WT ADE2 sequence. The resulting strain (GY2752) was then transformed with a 4Kb *Not*I fragment from pGY472.

For ChIP, strain GY2416 containing a single *loxP* site at the *LYS2* locus on the right arm of chromosome 2 was obtained by transforming strain GY855 with plasmid pGY621 linearized at *Nru*I followed by pop-in and pop-out selections as above.

## Purification of recombinant Cre and LiCre from bacteria

The coding sequences of Cre and LiCre fused to a N-terminal 6xHIS tag were codon-optimized for production in *Escherichia coli*. They were synthesized by GeneCust and cloned in a pET-19 vector. The resulting plasmids (pGY611 and pGY709) were transformed into Rosetta 2 bacteria. Bacterial cells were cultured in LB medium containing ampicillin and chloramphenicol at 37°C. A 40 mL starter culture was used to inoculate a 1 L culture. When reaching $OD_{600}$=0.6, these cultures were induced for 3 h with 0.1 mM IPTG. Cells were harvested by centrifugation and resuspended in 50 mL of 50 mM Tris (pH 8), 500 mM NaCl, 10% glycerol, 0.1% Triton, 0.5 M glucose, 1 mM DTT, 1 g/L lysozyme, 2× mix of protease inhibitors, 5 μg/mL DNase and 2.5 mM MgCl$_2$. After 30 min on ice, the suspension was sonicated four times for 1 min and then centrifuged at 14,000 *g* for 20 min at 4°C. The resulting supernatant was filtered on a GD/X membrane (INI) and then loaded on a HisTrap 1 mL column (GE Healthcare) using an AKTA PURE chromatography system and a varying ratio of two buffers, A and B. Buffer A contained 50 mM Tris (pH 8), 500 mM NaCl, 10% glycerol. Buffer B had the same composition as Buffer A with imidazole added at 300 mM. Three series of injections and washes were performed using A:B mixtures of different proportions: 100%:0%, 95%:5% and 90%:10%. Elution was done by applying a gradient of A:B mixtures from 90%:10% to 50%:50%, which lasted 10 CV (column volume, taking into account the packing and porosity of the material), followed by a 0%:100% mixture for 5 CV. Fractions were analyzed by SDS-PAGE. We noted that the eluted fractions of interest had a yellow color (Fig. S2a), indicating that LiCre was co-purified with its cofactor FMN. Fractions of interest were concentrated on a 10 kDa membrane (Vivaspin, Millipore). The resulting products were purified through two columns: a GF Superdex 75 10/300 Increase column (Cytiva) and a Zeba column, which was eluted in the final

SPR buffer 20 mM Tris (pH 7.4), KCl 50 mM, NaCl 150 mM, MgCl$_2$ 10 mM, DTT 1 mM, P20 surfactant 0.05%. The quality of the production was assessed by light diffusion on a NanosizerS (Malvern) and SDS-PAGE (Fig. S2b).

## ChIP

Strains GY2416 and GY2758 (negative control) were transformed with plasmid pGY44, pGY605, pGY607 or pGY820. Three independent transformants were used as biological replicates (nine strains in total). For each replicate, the strain was cultured overnight in 120 mL liquid flasks of synthetic SD-MW medium containing 2% glucose. The composition of this medium was the same as for the SD-Met medium previously described (Ansel et al., 2008), except that tryptophan was missing in the dropout mix. The next day, 54 mL of each culture was loaded on a six-well plate (9 mL per well) and illuminated in a DMX-controlled 450 nm-LED box placed in a 30°C incubator. Light intensity was set to 35 mW/cm$^2$. Illumination consisted of periodic pulses of 1 s applied every 10 s for a total duration of 60 min. In parallel, 54 mL of the same culture was kept in the dark. For each sample (illuminated or not), 25 mL was processed for WB and 25 mL was processed for ChIP. For WB, cells were washed twice with TBS1×, aliquoted in microtubes at $10^8$ cells per tube and pelleted by centrifugation. For ChIP, 700 μL of 37% formaldehyde (Sigma, F8775) was added to the 25 mL cell suspension. After 15 min at room temperature, 1250 μL of 5 M glycine was added and tubes were placed on a rotating wheel for 5 min at room temperature. Cells were then washed twice with TBS1×, aliquoted in microtubes at $10^8$ cells per tube and pelleted by centrifugation. All cell pellets (WB and ChIP) were frozen in liquid nitrogen and stored at −80°C.

For WB, cell pellets were thawed on ice and resuspended in 300 μL lysis buffer composed of 0.1% w/v sodium deoxycholate, 1 mM EDTA (pH 8), 50 mM HEPES-KOH (pH 7.5), 140 mM NaCl, 1% w/v Triton X-100 and proteinase inhibitors (Sigma, P2815), mixed by pipetting and transferred to Sarstedt tubes (ref. 72.693.005). A ~500 μL suspension of cold 'acid-washed' beads (Sigma, G8772) was added, and the tubes were shaken in a PreCellys machine (Bertin Technologies) at 6800 rpm during two cycles, each cycle comprising 10 s ON, 30 s OFF and 10 s ON. Tubes were placed on ice for 5 min between the two cycles. The lysate was recovered by piercing the bottom of the tube with a heated needle, placing it in a hemolysis tube, and spinning for 1 min at 2500 rpm and 4°C. Total protein amounts in the recovered lysates were quantified using the Bradford-based Bio-Rad Protein Assay (#500-0006), and samples were adjusted to contain 50 μg of protein in 12 μL. Prior to electrophoresis, 2.5 μL of loading mix 5× [0.5 M Tris (pH 6.8), 32% glycerol, 8% SDS, 8% DTT, 0.03% Bromophenol Blue] was added, and tubes were incubated for 5 min at 95°C. Proteins were separated by SDS-PAGE on a 12% acrylamide gel and transferred in semidry conditions to a 0.45 μm nitrocellulose membrane, which was then washed in TBS+Tween 0.02% for 10-15 min at room temperature, blocked by incubation in TBS+5% milk for 1 h and washed again three times for 10 min in TBS+Tween 0.02%. To quantify relative loadings, we incubated the membrane with an anti-GAPDH primary antibody (GA1R, Thermo Fisher MA5-15738) diluted at 1/5000, washed it three times for 10 min in TBS+Tween 0.02%, incubated it with HRP-coupled secondary antibody (GE Healthcare NA9310) diluted at 1/5000, washed it three times for 10 min in TBS+Tween 0.02%, and revealed it by the Enhanced chemiluminescence (ECL) reaction (Cytiva RPN 2232), which was quantified on a ChemiDoc reader (Bio-Rad). To quantify the LiCre-V5 and Cre-V5 proteins, the membrane was washed in TBS+Tween 0.02%, then incubated with a diluted anti-V5 primary antibody (Bio-Rad MCA1360) at a dilution of 1/1000 for 2 h at room temperature. The membrane was then washed three times for 10 min in TBS+Tween 0.02%, incubated with a diluted HRP-coupled secondary antibody (Cytiva NA9310) at a dilution of 1/5000, and washed three times for 10 min in TBS+Tween 0.02%. The membrane was then processed again for ECL and quantification. Our study included positive and negative controls that validated the use of these antibodies (Fig. 1; Fig. S1).

For ChIP, aliquoted fixed cells were thawed on ice and resuspended in 300 μL lysis buffer composed of 0.1% w/v sodium deoxycholate, 1 mM EDTA (pH 8), 50 mM HEPES-KOH (pH 7.5), 140 mM NaCl, 1% w/v Triton X-100 and proteinase inhibitors (Sigma, P2815). Tubes were mixed by

pipetting. To disrupt cells, we added 500 µL of cold acid-washed glass beads (Sigma, G8772) and placed the tubes on a PreCellys apparatus. Tubes were shaken at 6800 rpm through two cycles, each cycle consisting of 10 s ON, 10 s OFF, 10 s ON. Tubes were placed for 1 min on ice between the two cycles. Cell lysates were recovered and sonicated on a Covaris apparatus with the following settings: P 140 W, duty factor 5%, 200 bursts per cycle, 15 min. Debris were eliminated by centrifugation at 14,000 rpm and 4°C. We then added lysis buffer to reach a volume of 1 mL. After vortexing, 200 µL was used to control sonication efficiency on agarose gels, 750 µL was used for immunoprecipitation (IP) and 50 µL was kept as 'inputs'. For IP, anti-V5 monoclonal antibody (Bio-Rad MCA1360) was captured on Dynabeads pA (Life Technologies 10002D), and 35 µL of this preparation was added to the sample; tubes were placed overnight at 4°C on a rotating wheel. In parallel, the 'input' samples were also placed overnight at 4°C. The following day, 50 µL of lysis buffer was added to the 'input' samples, followed by 35 µL of naked (i.e. without antibody) pA Dynabeads, and this mixture was incubated for 1 h at 4°C on a rotating wheel. For IP and input samples, beads were washed three times in 0.5 mL of washing buffer I [ 20 mM Tris-HCl (pH 8), 150 mM NaCl, 2 mM EDTA, 1% Triton X-100, 0.1% SDS], once in 0.5 mL of washing buffer II [20 mM Tris-HCl (pH 8), 500 mM NaCl, 2 mM EDTA, 1% Triton X-100, 0.1% SDS], once in 0.5 mL of washing buffer III [10 mM Tris-HCl (pH 8), 1 mM EDTA, 0.5% deoxycholate, 1% IGEPAL, 25 mM LiCl] and twice in 0.5 mL of TE8 buffer [10 mM Tris-HCl, 1 mM EDTA (pH 8)]. For elution, beads were resuspended in 130 µL of elution buffer [30 mM Tris-HCl (pH 8), 2 mM EDTA, 1% SDS, 0.35 mg/ml proteinase K], incubated at 65°C for 5-6 h and briefly vortexed every hour. Eluates were transferred to fresh tubes, and beads were further incubated in 20 µL of 1 M Tris (pH 5.5) to generate a second eluate that was also transferred and added to the first. DNA was further purified on NucleoSpin® Gel and PCR Clean-up Columns (Macherey-Nagel) and finally eluted in 23 µL of ultrapure water. Real-time qPCRs were run on a Rotor-Gene Q thermocycler from QIAGEN using primer pairs (1R69,1R70), (1R71,1R72), and (1R73,1R74), corresponding to probes mentioned in Fig. 1C as P1, P2 and P3, respectively. Unrelated control loci were quantified using primer pairs (1H97,1H98) located in *HIS5*, (1E10,1E11) located in *AHA1* and (1B36,1B37) located in *iSWH1*. We estimated relative quantification (ChIP/INPUT) by computing the NRQ ratio described by Hellemans et al. (2007). This ratio follows $NRQ=(E^{\Delta Ct})/R$, where $E$ is the amplification efficiency of the target amplicon of interest (for example, P1), $\Delta Ct$ is the difference in $Ct$ values for this amplicon between ChIP and INPUT, and $R$ is the geometric mean of three reference $E_i^{\Delta Cti}$ values, with $E_i$ being the amplification efficiency of reference $i$ and $\Delta Cti$ being the difference in $Ct$ values for reference $i$ between ChIP and INPUT. This way, every ratio was normalized by all three references $i \in \{HIS5, AHA1, iSWH1\}$.

## Analysis by SPR

We conducted experiments on a Biacore™ T200 (Cytiva) apparatus equipped with Biacore T200 control software v3.2.1, using streptavidin-coated Series S Sensor Chips SA (Cytiva). We conditioned the chip according to the manufacturer's recommendations (three consecutive 1-min injections of 1 M NaCl in 50 mM NaOH). After conditioning, we used, for all experiments, a running buffer composed of 20 mM Tris (pH 7.4), 50 mM KCl, 150 mM NaCl, 10 mM MgCl₂, 1 mM DTT, 0.05% P20 surfactant (Cytiva). The buffer was filtered on a 0.22 µm MCE membrane (Millipore) before use. Recorded signals were expressed in RUs corresponding to a rise in SPR signal resulting from protein/DNA interactions. We prepared ligands by annealing complementary oligonucleotides corresponding to full *loxP* (1Y68/1Y69) or half *loxP* (1Y70/1Y71) or a random sequence (1Y72/1Y73). For each pair, one of the oligonucleotides contained a biotin group at its 5′ end (Table S3). Each sensor chip contained four flow cells (channels). We first used these channels independently to capture DNA ligands: 32.1 RU of control DNA in channel 1 of the chip, 31.6 RU of half *loxP* DNA in channel 2, 10.6 RU of control DNA in channel 3 and 10.8 RU of *loxP* DNA in channel 4. For kinetic measurements, we then used a series of two consecutive channels: FC2-1 (half *loxP*) and FC4-3 (full *loxP*). The temperature of the chip was set at 25°C. Samples were kept in the Biacore chamber at 10°C. The flow rate was set at 100 µL/min. To measure association, we injected Cre or LiCre for 90 s at concentrations varying from 0.3125 to 10 nM prepared as serial twofold dilutions. We then monitored

dissociation for 400 s. We applied a regeneration step between each sample to ensure that all detectable Cre or LiCre molecules were dissociated from DNA. These steps consisted of two 30-s injections of 2 M MgCl₂ in 0.5 M NaCl for full *loxP* kinetics and one 60-s injection of 2 M NaCl for half *loxP* kinetics. Sensorgram artifacts were corrected by subtracting a control injection of running buffer.

## Kinetic model of protein:DNA binding

We based this model on the previous work of Rüfer et al. (2002). For binding to a half-*loxP* site, we assume that the binding of Cre or LiCre is defined by the association ($k_1$) and dissociation ($k_{-1}$) rates and follows the mass-action model (Fig. 3A):

$$\frac{d[P_1]}{dt} = k_1[C]([P_{tot}]-[P_1]) - k_{-1}[P_1], \tag{1}$$

with $[P_{tot}]$ being the total concentration of half-*loxP* sites, $[P_1]$ being the concentration of occupied half-*loxP* sites and $[C]$ being the concentration of Cre/LiCre units. In the SPR protocol, Cre or LiCre proteins are injected at a controlled concentration $[C_i]$. They diffuse within the biosensor chip and associate with half-*loxP* sites attached to the surface (Fig. 2). This leads to the following equation of evolution for $[C]$:

$$\frac{d[C]}{dt} = k_t([C_i]-[C]) - k_1[C]([P_{tot}]-[P_1]) + k_{-1}[P_1], \tag{2}$$

with $k_t$ as the coefficient of mass transport within the device.

For binding to a full *loxP* site, we assume a two-step process (Fig. 3A): one unit first binds with the same kinetic rates as for the half-*loxP* site ($k_1$, $k_{-1}$), the second unit then associates with possibly different kinetics ($k_2$, $k_{-2}$). This way, putative cooperative effects can be captured. This leads to the following mass-action model:

$$\frac{d[C]}{dt} = k_t([C_i]-[C]) - 2k_1[C]([P_{tot}]-[P_1]) + k_{-1}[P_1] - k_2[C][P_1] + 2k_{-2}[P_2], \tag{3}$$

$$\frac{d[P_1]}{dt} = 2k_1[C]([P_{tot}]-[P_1]) - k_{-1}[P_1] - k_2[C][P_1] + 2k_{-2}[P_2], \tag{4}$$

$$\frac{d[P_2]}{dt} = k_2[C][P_1] - 2k_{-2}[P_2], \tag{5}$$

with $[P_1]$ as the concentration of half-occupied *loxP* sites and $[P_2]$ as the concentration of fully occupied sites.

The output signal $[B](t)$ of SPR experiments, given in RUs, effectively measures the total mass of Cre/LiCre units bound to DNA sites at a given time $t$ and is thus proportional to $[P_1]$ in the half-*loxP* experiments, noted ($[B]^h(t)\equiv\alpha[P_1]^h$) with superindex $h$ and to $[P_1]+2[P_2]$ in the full-*loxP* experiments, noted ($[B]^f(t)\equiv\alpha([P_1]^f+2[P_2]^f)$) with superindex $f$. The maximal values for $[B](t)$ are termed $RU_{max}$ and are defined as $RU_{max}^h = \alpha[P_{tot}]^h$ and $RU_{max}^f = 2\alpha[P_{tot}]^f$. Each experiment consists of two successive steps: an association phase (duration $t_a$) where the protein is injected ($[C_i]\equiv[C_0]>0$ for $0\leq t\leq t_a$) and a dissociation phase in the absence of new injected material ($[C_i]=0$ for $t\geq t_a$). To quantitatively compare model predictions with experiments, for a given experimental condition (Cre or LiCre, half- or full-*loxP* site, $[C_0]$) and a given set of model parameters ($k_1$, $k_{-1}$, $k_2$, $k_{-2}$, $k_t$, $\alpha$, $[P_{tot}]$), we numerically solved the systems of ordinary differential equations Eqns 1 and 2 or 3-5 with initial conditions $\{[C],[P_1],[P_2]\}=\{0, 0, 0\}$ and then computed the corresponding SPR-like output signal (see above).

We fitted model parameters as follows. For each type of molecule (Cre or LiCre) and site (half- or full-*loxP*), experimental data are available for seven values of $[C_0]$ and a fixed $t_a$ value of 90 s (Fig. 2). For a given set of parameters ($k_1$, $k_{-1}$, $k_2$, $k_{-2}$, $k_t$, $\alpha$, $[P_{tot}]$), we simulated all experimental conditions ($n=14$) and computed a chi-square-like score defined as follows:

$$\chi_2 = \sum_{e,d} \left[\frac{y_{d,e}-f(t_d;param)}{y_{d,e}+\varepsilon}\right]^2, \tag{6}$$

with $y_{d,e}$ as the SPR signal observed at time $t_d$ ($\in[0:0.1:340]sec$) during experiment $e$, $f(t_d;param)$ as the corresponding model prediction and $\varepsilon=10$

as a meta-parameter that grossly estimates the data uncertainty. Note that one score was defined for the Cre dataset and another for LiCre. The parameters for each type of molecule were fitted independently by minimizing the corresponding $\chi_2$ using a trust region reflective algorithm (function *least_squares* in SciPy) starting from initial parameter values randomly sampled within acceptable ranges (Table S4). We repeated the minimization 1000 times per type of molecule and stored, for each fitting round, the optimal set of parameters and the minimum score. The distribution of minimum scores showed that some rounds of minimization reached only a local minimum and did not fully optimize the fit (Figs 2A-D, 3B,C). We discarded them from the analysis by setting an arbitrary threshold above the first mode of the distribution, keeping only 541 and 859 sets of fitted parameters for Cre and LiCre, respectively.

### Quantification of LiCre-*loxP* recombination in yeast
The yeast assay presented here was performed as previously described (Duplus-Bottin et al., 2021). Yeast reporter strains were transformed with the $P_{met17}$-LiCre plasmid pGY466 or plasmid pGY44 (empty vector) or $P_{met17}$-Cre plasmid pGY502 (positive control). For each strain-plasmid combination, three independent transformants were used as biological replicates. For each replicate, a fresh colony was used to inoculate 4 ml of synthetic selective medium lacking methionine (in order to induce LiCre expression) with no particular protection against ambient light. After 18 h, the saturated culture was transferred to two 96-well polystyrene flat-bottom sterile plates (100 µl per well); one plate was illuminated with the indicated conditions while the other plate was kept in the dark at the same temperature. We used two devices for photoactivation. The first device was a 50 W spot (Neptune-LED, Grenoble, France) equipped with a 450 nm LED connected to a DMX controller, which we placed above a thermostated platform. The second device was a homemade aluminum box equipped (by Neptune-LED, Grenoble, France) with a ceiling of LED strips (450 nm) connected to a DMX controller. This box was placed into an incubator at 30°C. Both devices were controlled from a laptop computer via the QLC+ software. After illumination, cells from the two plates were diluted into fresh synthetic medium in deep-well plates and incubated at 30°C for 4 h to allow expression of the fluorescent protein. We then analyzed cells by flow cytometry either immediately or the day after. In this latter case, cells were either kept in PBS+1 mM sodium azide or fixed with 2% paraformaldehyde (PFA) before they were analyzed the following day. We then analyzed the raw flow cytometry data using custom R scripts as previously described (Duplus-Bottin et al., 2021). No statistical power analysis was done prior to the experiment, no masking of sample IDs was applied during the experiments, and no data randomization was used.

### Statistical tests
No power analysis was performed: experimental sample sizes were chosen to be of at least three independent biological replicates per condition. Pairwise tests presented in figures were made in R v. 4.5.2 using the *t-test*() function, which applies a Welch two-sample *t*-test that accounts for inequality of variances when necessary. The experimenter was not blinded to the identity of the samples.

### Kinetic model of recombination
For recombination to happen, we assume that (i) both *loxP* sites should be fully occupied by two LiCre units each and that (ii) at least $x$ among four LiCre units should be currently in the activated state. If we neglect the backward, reinsertion reaction (see main text), the probability $p$ for a cell to have not recombined yet follows the evolution equation:

$$\frac{dp(t)}{dt} = -R(t)p(t) \quad \text{with} \quad R(t) = R_0 P_{2,2}(t) A_x(t), \qquad (7)$$

where $P_{2,2}(t)$ is the probability at a given time $t$ that condition (i) is effective, $A_x(t)$ is the probability that condition (ii) is realized and $R_0$ is the recombination rate when both conditions (i) and (ii) are fulfilled. Note that $R_0$ encompasses the rate of DNA looping between the two *loxP* sites, the rate of synapse formation and the actual rate of recombination when the synapse is formed and functional. From Eqn 7, the fraction $\phi(t)$ of cells having

recombined at time $t$ is given by:

$$\Phi(t) \equiv 1 - p(t) = 1 - exp\left(-\int_0^t dt' R(t')\right), \qquad (8)$$

where we assumed that initially ($t$=0) no cells have recombined ($p(0)$=1).

### Occupation of *loxP* sites
We assume that the binding of LiCre at one *loxP* is independent from binding events at the other *loxP* site, which we write as follows:

$$P_{2,2}(t) = P_2(t)^2, \qquad (9)$$

with $P_2(t)$ as the probability that one site is fully occupied, which is given by Eqns 4 and 5 by replacing $[P_1]$ and $[P_2]$ by $P_1(t)$ and $P_2(t)$ (the probabilities that the site is occupied by one and two LiCre units, respectively) and by setting $[C]$ as a constant equal to the average nuclear concentration of LiCre.

Since light does not impact LiCre binding (Fig. 1) and binding/unbinding of LiCre is much faster (approximately seconds, Fig. 3F) than the recombination step (minutes), we assume that occupation of *loxP* sites is at steady state. Finding the stationary solutions of Eqns 4 and 5 and using Eqn 9 lead to:

$$P_{2,2} = \left(\frac{[C]^2}{[C]^2 + 2K_{d2}[C] + K_{d1}K_{d2}}\right)^2. \qquad (10)$$

### Photoactivation of individual LiCre units
Assuming that the activation of one LiCre unit is independent from the state of the three other units bound to the two *loxP* sites:

$$A_x(t) = \begin{cases} (1 - (1 - P_{act}(t))^4) & \text{for} \quad x = 1 \\ P_{act}^4(t) + 4P_{act}^3(t)(1 - P_{act}(t)) \\ \quad + 6P_{act}^2(t)(1 - P_{act}(t))^2 & \text{for} \quad x = 2 \\ P_{act}^4(t) + 4P_{act}^3(t)(1 - P_{act}(t)) & \text{for} \quad x = 3 \\ P_{act}^4(t) & \text{for} \quad x = 4 \end{cases} , \qquad (11)$$

with $P_{act}(t)$ being the probability that one LiCre unit is activated. The kinetics of activation/inactivation is supposed to follow a simple two-state model:

$$\frac{dP_{act}}{dt} = k_{ON}\Delta(t)(1 - P_{act}) - k_{OFF}P_{act}, \qquad (12)$$

where $k_{ON}$ is the activation rate when light is ON and depends on the lighting intensity (see below), $\Delta(t)$ is a time-dependent binary variable that is equal to 1 when the light is ON and 0 otherwise, $k_{OFF}$ is the deactivation rate. For a periodic illumination defined by a period $T$ and a duty cycle $dc$ that represents the fraction of $T$ during which the light is ON, Eqn 12 can be solved analytically:

$$\begin{aligned} P_{act}(t) = \left(\frac{k_{ON}}{k_{ON} + k_{OFF}}\right)&\Big\{\Theta(dc - (t/T - n))((1 - e^{-(k_{ON}+k_{OFF})(t-nT)}) \\ &+ (1 - e^{-(k_{ON}+k_{OFF})dcT})e^{-k_{OFF}(1-dc)T} \\ &\times \left(\frac{1 - e^{-nT(k_{ON}dc+k_{OFF})}}{1 - e^{-T(k_{ON}dc+k_{OFF})}}\right)e^{-(k_{ON}+k_{OFF})(t-nT)}\Big) \\ &+ \Theta((t/T - n) - dc)(1 - e^{-(k_{ON}+k_{OFF})dcT}) \\ &\times \left(\frac{1 - e^{-(n+1)T(k_{ON}dc+k_{OFF})}}{1 - e^{-T(k_{ON}dc+k_{OFF})}}\right)e^{-k_{OFF}(t-nT-dcT)}\Big\}, \end{aligned} \qquad (13)$$

with $n$=$t/T$ and $\Theta(u)$ being the Heaviside function.

### Choice of parameters, integration and fitting
For a given set of parameters ($R_0$, $K_{d1}$, $K_{d2}$, $[C]$, $k_{ON}$, $k_{OFF}$, $T$, $dc$), the time evolution of $\phi(t)$ can be computed using Eqns 7, 8, 10, 11 and 13. To

compare it to experimental data, we fix parameters for which a value was known. We chose the $K_{d1}$, $K_{d2}$ values that were inferred from SPR experiments on LiCre (Fig. 3F). For $k_{ON}$, we used the estimate of Renicke et al. (2013), who computed the activation rate of a LOV2 domain as $k_{ON}=0.26SF$ with $S=4.3\times10^{-17}$ cm$^2$ as the surface of LOV2, $F=(E \lambda 10^{13})/2$ cm$^{-2}$ s$^{-1}$ as the flux of incoming photons, $\lambda$ as the light wavelength (in $nm$), and $E$ as the power of the light in mW·cm$^{-2}$. In our case, this led to $k_{ON}\approx(E/35)s^{-1}$. The concentration of LiCre in the yeast nucleus is not known. For simplicity, we fixed it at $[C]$=1 nM, which represents about 100 molecules per nucleus. $E$, $T$ and $dc$ are defined by the illumination conditions of each experiment.

To fit the remaining parameters ($x$, $R_0$ and $k_{OFF}$), we considered four sets of experiments listed in Table S5, each set corresponding to a coherent series (proximal dates, same operator and comparable conditions): experiment 8 (Fig. 5D), experiments 1 and 2 (Fig. 5E), experiments 3 to 7 (Fig. 5F) and experiment 9 (Fig. 5G). For each set of experiments, we systematically varied $x$ from 1 to 4, $R_0$ between $5.10^{-5}$ and $1.5$ $10^{-3}$ s$^{-1}$ and $k_{OFF}$ between $8.10^{-4}$ and $2.5$ $10^{-2}$ s$^{-1}$, we computed the corresponding model prediction for $\phi(t)$, and we estimated a chi-square-like score (similar to Eqn 6) where experimentally measured recombination efficiencies were normalized by a factor of 0.87 to account for the frequency of LiCre-plasmid loss (see main text). For each set $e$ of experiments and each $x$ and $k_{OFF}$ values, we estimated the minimum chi-square-like score $\chi_2^{min}(e; k_{OFF}, x)$ and the corresponding $R_0(e)$ value. We then estimated the value of $x$ and $k_{OFF}$ that minimized $\sum \chi_2^{min}(e; k_{OFF}, x)$. For each $x$ value, optimal $\{R_0(e;x)\}$ and $k_{OFF}(x)$ values are shown in Table 1. Note that we fitted the model by imposing the same values of $x$ and $k_{OFF}$ for every experiment. In contrast, $R_0$ was estimated for each experiment in order to capture the observed batch-to-batch experimental variability (which was presumably due to changing environmental conditions such as temperature, see main text). Note also that our arbitrary choice for $[C]$ does not impact the quality of the fit nor the best values for $x$ and $k_{OFF}$ because changing its value changes only $P_{2,2}$ (Eqn 10), which acts as a normalizing factor in Eqn 7. Nonetheless, the choice of $[C]$ may impact the fitted values of $R_0(e;x)$ as follows:

$$R_0(e; x; [C]) = R_0(e; x; [C] = 1\,\text{nM}) \frac{P_{2,2}([C] = 1\,\text{nM})}{P_{2,2}([C])}, \qquad (14)$$

which has a rather slow dependency in $[C]$, and thus, the fitted value would be rather similar unless $[C] \ll 1$ nM, which is unlikely (see Discussion).

To separately estimate $k_{OFF}$ and $R_0$ in WT LiCre and in the T418S mutant, we applied a similar strategy as described above but using only Dataset 13 (Table S5). For each variant (WT or T418S) and each $x$ value, we computed a chi-square score over a 2D grid of parameters for $k_{OFF}$ and $R_0$, accounting for all the experimental data of the variant. Parameters that minimize this score are given in Table S7.

### Relation between $R_0$ and kinetic parameters previously estimated for Cre

Shoura et al. (2012) fitted *in vitro* FRET measurements by a three-state model of intramolecular Cre-*loxP* recombination (figure 6A of their publication). They estimated the rate of synapse formation from the unlooped state $k_3$, the rate of synapse unfolding $k_{-3}$ and the rate of recombination catalysis $k_{-4}$. They found that $k_{-4}$ was the rate-limiting step in the two cases investigated (inter-*loxP* distances of 3044 and 870 bp). In our model, $R_0$ encompasses all these different reactions. In the limit $k_{-4}=k_3$, $k_{-3}$, we can approximate $R_0$ by the probability for a non-edited DNA molecule to be in the synaptic state, multiplied by the catalysis rate, giving $R_0=k_{-4}(k_3)/(k_3+k_{-3})$. We used this formula to compare our $R_0$ values with corresponding estimates from table 2 of Shoura et al. (2012) (see Discussion).

### Intracellular LiCre concentration

The effective concentration of 'free' recombinase available to conduct DNA recombination in live cells is unknown. It is, however, related to the average intracellular concentration of LiCre, which we anticipate is largely above 1 nM for the following reasons. In our conditions, LiCre expression was driven by the P$_{MET17}$ promoter and its level is therefore likely comparable to the level of Met17p. In a previous study based on LC-MS, this protein was

estimated at ~120,000 copies per yeast cell (equivalent to 4 μM if we assume a cell volume of 50 μm$^3$), an abundance 1.6 times higher than the average abundance of the proteins quantified in this study (data S1B of Martin-Perez and Villén, 2017). Another study based on fluorescently tagged proteins (Ghaemmaghami et al., 2003) estimated an average concentration of 12,100 copies per cell (0.4 μM) per protein. So unless LiCre is particularly unstable in yeast, its concentration should be two or three orders of magnitude above 1 nM.

### Acknowledgements

This work has benefited from the expertise of the Protein Science (Virginie Gueguen-Chaignon and Aurélie Thibaut, who adjusted conditions for protein purification) and Flow Cytometry facilities of the SFR Biosciences Lyon, who trained authors, provided technical support and access to equipments. We thank Fabien Duveau, Mirko Francesconi and Aurèle Piazza for fruitful discussions; Fabien Duveau, Mirko Francesconi and Valérie Robert for critical reading of the manuscript; five anonymous reviewers for their comments; Pascal Bernard for a plasmid carrying V5-epitope repeats; Jean-Philippe Robin for advice on recombinant proteins; and developers of R, Bioconductor, QLC+, NumPy, Git and Ubuntu for their software.

### Competing interests

The authors declare no competing or financial interests.

### Author contributions

Conceptualization: D.J., G.Y.; Data curation: A. Dufour, G.Y.; Formal analysis: A. Dufour, H.D.-B., G.T., D.J., G.Y.; Funding acquisition: D.J., G.Y.; Investigation: A. Dufour, H.D.-B., G.T., C.D.-K., A. Dumont, D.J.; Methodology: A. Dufour, H.D.-B., T.B.-L., E.C., G.T., C.M., D.J., F.V., G.Y.; Project administration: G.Y.; Resources: C.M., G.Y.; Software: T.B.-L., E.C., L.T., D.J.; Supervision: D.J., G.Y.; Validation: H.D.-B.; Visualization: A. Dufour, T.B.-L., E.C., L.T., D.J., G.Y.; Writing – original draft: G.Y.; Writing – review & editing: D.J., G.Y.

### Funding

This work was supported by the Centre National de la Recherche Scientifique (CNRS) under the 'MITI 80 Prime' program grant READGEN; the Fondation ARC pour la recherche contre le cancer; the Agence Nationale de la Recherche Grant (ANR) Opto4D ANR-23-CE12-0039; funding for interdisciplinary innovations from the Institut RhôneAlpin des Systèmes Complexes (IXXI) and the Federation for Systems Biology in Lyon (BioSyL); and funding for technological developments from the SFR Biosciences Lyon. Open Access funding provided by CNRS. Deposited in PMC for immediate release.

### Data and resource availability

Supplementary Data contain the source data of all figures (supporting Dataset), computational codes implementing the models of LiCre:DNA binding and recombination reaction, which are also made available at https://github.com/physical-biology-of-chromatin/licre-optogenetic-kinetics/, as well as Supplementary Tables and Figures.

### Peer review history

The peer review history is available online at https://journals.biologists.com/bio/lookup/doi/10.1242/bio.062381.reviewer-comments.pdf

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
