## [Peer Review File · Biology Open]

Kinetic properties of optogenetic site-specific DNA recombination by LiCre-loxP

Alice Dufour, H el ene Duplus-Bottin, Thomas Boukeke-Lesplulier, Eliane Casassa, Gerard Triqueneaux, Leo Tarbouriech, Camille Darthenay-Kiennemann, Agnes Dumont, Catherine Moali, Daniel Jost and Ga el Yvert

DOI: 10.1242/bio.062381

Editor: Alissa Armstrong

Review timeline

Original submission: 19 November 2025

Editorial decision: 21 November 2025

First revision received: 19 February 2026

Accepted: 3 March 2026

Reviewer 1:

Evidence, reproducibility and clarity

Dufour et al describe characterization of the light-activated recombinase LiCre. This work combines the yeast reporter assay, surface plasmon resonance (SPR) and kinetic modeling to provide a comprehensive study of how LiCre functions both in vivo in yeast and in vitro. The authors show that LiCre binds to loxP sites in the dark with high affinity, but reduced cooperativity compared to wild-type Cre, and that recombination efficiency is affected by temperature and illumination regime. Importantly, the authors establish a kinetic model that not only explains these observations but also predicts the altered behavior of a mutant (T418S), which was experimentally validated. It would be valuable to highlight what other predictions the model could make, even if for future work. Overall, this work combines quantitative experiments and modeling to provide new insights into the biochemical and kinetic properties of LiCre.

Specific comments:

Line 110-115: Although described in the Methods section, a brief statement of dark and light treatment conditions would help readers better follow the experiments. Likewise, listing the three unrelated positions would improve the clarity.

Line 185: Is there a typo?

Line 216: Have the authors considered performing surface plasmon resonance (SPR) to confirm the binding affinity of LiCre-V5 DNA?

Line 233-234: To determine whether the observed difference in recombination efficiency is due to the genomic context of the reporter loci or due to the measurement accuracy of GFP and RFP signals, have the authors considered swapping the positions of GFP and RFP?

Line 236: The sentence "Importantly, we never observed recombination in the entire cell population" is ambiguous. I believe it means recombination was never observed in 100% of the cells. Please rephrase it.

Line 245-249: The hypothesis of plasmid loss based on plating samples on selective and non-selective media without illumination assumes that loss of growth on selective media is only due to plasmid loss, without considering other factors like burden or toxicity. Moreover, the broad range

of 10-30% makes it difficult to justify that the ~15% recombination-negative fraction falls within expected variation. The conclusion that LiCre-mediated recombination efficiency is close to 100% after prolonged photoactivation (Line 249, 301-303) is not fully convincing unless more evidence is provided.

Line 275-276: The authors suspect that the decrease in recombination efficiency at very high light intensity is possibly attributed to phototoxicity. Could photobleaching also contribute to this effect? A viability assay would help to validate the phototoxicity explanation.

Line 345-346: While the model with $x=2$ provides a slightly better fit comparing to the others, the possibility of $x=4$ cannot be excluded. The inference that "photo-activation of at least two LiCre protomers enables recombination" is not sufficiently proven.

Figure 1e: Please clarify whether the Western blots shown represent biological replicates.

Figure 4: Please include the error bars. Panel a - The authors integrated GFP and mCherry reporters at two different loci to avoid positional bias. Why then is only mCherry used as the ON readout in most experiments, rather than analyzing both reporters in parallel? Please clarify. For panel 4h and line 272, the statement that maximal activation was reached at 12 mW/cm² should be rephrased more cautiously, as no intermediate intensities between 12 and 35.6 mW/cm² were tested.

Significance

This study provides a quantitative experimental and predictive analysis of the light-activated recombinase LiCre, offering new insights into its binding, activation and recombination properties. The predictive validation of the mutant is a strength of this work. While the modeling part is an innovative aspect, more clarification is needed, especially regarding the conclusion that photo-activation of at least two LiCre protomers enables recombination. More mechanistic investigations are needed to support the conclusions. The work will be of interest to researchers in optogenetics, genome engineering, and DNA-protein interactions. My expertise is in yeast genome engineering and applications of Cre-mediated recombination system. Modeling is outside my primary area of expertise.

Reviewer 2:

Evidence, reproducibility and clarity

****Summary:**** This manuscript presents a detailed kinetic and mechanistic characterization of the optogenetic recombinase LiCre, which enables site-specific DNA recombination upon blue-light stimulation. The authors combine in vitro surface plasmon resonance assays, yeast-based recombination assays, and mathematical modeling to dissect the DNA-binding properties, activation dynamics, and recombination efficiency of LiCre. They demonstrate that LiCre binds DNA even in the absence of light, albeit with reduced cooperativity compared to Cre recombinase. Through kinetic modeling, they propose that activation of only two LiCre units may suffice for recombination. The study also evaluates the impact of point mutations in the LOV domain on LiCre's photocycle. The experimental methods are described in detail. Statistical analyses are appropriate and clearly reported.

****Major Comments:****

1. In Figure 1, control experiments with no loxP sequences (i.e. original strain) should be performed to demonstrate specific binding of Cre/LiCre to loxP sequence.
2. In Figure 2, the SPR experiments are robust and informative. However, the lack of measurement of DNA binding of light-activated LiCre is a notable gap, which will help understand whether the cooperativity of LiCre can be modulated by light. If it is difficult due to experimental conditions, there is lit-mimetic mutant of LOV2 (<https://www.nature.com/articles/nmeth.3926>).

Significance

General Assessment: This is a rigorous study that combines experimental and computational approaches to advance our understanding of LiCre-based optogenetic genome engineering. The strongest aspects are the integration of SPR data with kinetic modeling and the practical insights into LiCre's performance under various conditions. However, the other limitation is the lack of direct validation of some model predictions.

Advance: To the best of my knowledge, this is the first study to quantitatively model the activation dynamics of LiCre. The work extends previous findings on LiCre and provides new mechanistic and practical insights.

Audience: This study will be of interest to specialized audiences, particularly those developing or applying the LiCre system.

Reviewers' Field of Expertise: Protein engineering, Genome editing, Optogenetics, Cell Biology.

Limitations of Expertise: I do not have deep expertise in mathematical modeling.

Reviewer 3:

Evidence, reproducibility and clarity

****Summary:****

The manuscript by Dufour et al. is a follow-up on the groups' previous publication that introduced the photo-inducible Cre recombinase, LiCre. In the present work, the authors further characterize the properties and kinetics of their optogenetic switch. Initially, the authors show that light affects only LiCre-mediated recombination itself and not DNA binding. Following these observations, they measure and mathematically model LiCre kinetics demonstrating high efficiency in vivo and a surprising temperature sensitivity. Finally, Dufour et al. evaluate several mutations that affect the LOV photo-cycle and provide recommendation for LiCre applications.

The study thoroughly investigates various aspects of the function of LiCre, confirming some previously known characteristics (i.e. temperature-dependence of Cre activity and functionality of LOV-based optogenetic tools in yeast without co-factor supplementation), while providing new LiCre-specific insights (kinetics, light-independent DNA binding). Please note that the reviewer is no expert in mathematical modeling and cannot fully judge the methodological details of the models. While I have some concerns as listed below, I believe study should be well-suited for publication after a revision.

****Major comments:****

1. After completing the initial experiment, the authors discovered that their plasmids carry different numbers of V5 epitopes. I am wondering whether this was due to a recombination event happening during the experiment or whether the constructs were not sequence verified prior to use? In any case, an additional ChIP experiment using Cre and LiCre constructs with the identical number of tag-repeats will be necessary. The result, i.e. the strong reduction of DNA-binding of LiCre (which is close to the negative control), is quite remarkable given that LiCre is still considerably active and high DNA affinities were observed in SPR experiments. In light of these counterindications, identical experiment conditions for test and reference group become even more important.
2. The conclusion that DNA-binding of LiCre is completely light-independent is not entirely convincing to me. The differences between the light and dark conditions in Fig. 2d are indeed small, but the values for LiCre are almost on par with the vector control and therefore hard to interpret. Based on this experiment alone, one could even be inclined to argue that LiCre does not bind DNA at all (which is of course falsified by the later experiments), showing that the resolution of the corresponding dataset is too low to draw final conclusions. Light-independent

DNA binding should either be confirmed by a more sensitive method or the conclusion statements on this matter should be revised accordingly.

3. If I understand the explanations correctly, replicates and plotted data points refer to multiple samples (different colonies), that were handled in a single experiment, i.e. by one researcher at the same time/same day. As already mentioned by the authors in the main text, this workflow explains the considerable differences between some of the results in the present manuscript and an identical experiment in a previous publication by the same authors. Providing truly independent experiments (performed on different days) that are therefore independent towards variables such as the fluctuation in incubation temperature (which was the issue in the described experiments) will be crucial, at least for the key datasets.

****Minor comments:****

1. At the end of the Introduction, the authors mention that the interaction of the Cre heptamers was weakened via point mutations in LiCre. A short sentence about the engineering rationale behind this weakened interaction would help readers, who are not familiar with the author's prior work.
4. Fig. 2a-b depicts images relating to the purification procedure. These could be moved to the supplements as they don't provide any insight apart from the fact that the proteins were successfully purified.
5. The kinetic characterization was only performed for LiCre. Especially for scientists, who have worked with wildtype Cre before, a side-by-side comparison with wt Cre would be valuable to judge the loss in reaction speed that has to be expected when switching from Cre to LiCre.
6. The difference between the ChIP results and the SPR results is striking but not mentioned in the discussion section. Also, the statement: "Finally, our results have practical implications on experimental protocols employing LiCre. First, given its high affinity for loxP (Fig. 5b), over-expressing LiCre at high levels will probably not increase its efficiency." (line 502) refers only to the affinity but seems to ignore the low DNA-occupancy of LiCre observed in Fig. 2d. Adapting the discussion section accordingly would improve the manuscript.

Significance

General assessment and advance:

The present study provides a large set of experiments and analyses characterizing the optogenetic LiCre recombinase. In general, the study is well conceived and executed. Although some of my concerns listed above affect key aspects of the study, they should be straightforward to address. The manuscript is a follow-up study providing a more detailed characterization of an optogenetic tool previously developed by the same authors. Its novelty is therefore somewhat limited. While the study provides a rich body of additional data, many of the findings merely confirmed aspects that were to be expected based on the two proteins LiCre is built of (temperature-dependent activity of Cre, optogenetics in yeast w/o the need of co-factor supplementation, weaker DNA-affinity of the Cre fusion protein as compared to wildtype Cre). New insights are provided by the facts that (i) light only controls recombination but not DNA binding and (ii) light activation of only some promoters within the LiCre heptamer is likely to be sufficient to activate recombination. The former aspect is, however, not entirely evident from the results as described above.

Audience:

The study will be of interest for researchers focusing on inducible DNA recombination and especially relevant to those who plan to work with LiCre and can now rely on a more detailed and extended characterization compared to the original LiCre publication.

Author response to reviewers' comments

Manuscript number: RC-2025-03141

Corresponding authors:

1. General Statements [optional]

Thank you for processing our manuscript through peer-reviewing. We appreciate the reviewers' feedback and we are pleased to upload a preliminary revision of the manuscript together with this revision plan.

2. Description of the planned revisions

Insert here a point-by-point reply that explains what revisions, additional experimentations and analyses are planned to address the points raised by the referees.

Planned revisions are numbered PR1, PR2, etc.

Reviewer #1

Line 233-234: To determine whether the observed difference in recombination efficiency is due to the genomic context of the reporter loci or due to the measurement accuracy of GFP and RFP signals, have the authors considered swapping the positions of GFP and RFP?

PR1. We plan to do this experiment and compare the strains with swapped positions for any difference in recombination efficiency.

Figure 4: Please include the error bars. Panel a - The authors integrated GFP and mCherry reporters at two different loci to avoid positional bias. Why then is only mCherry used as the ON readout in most experiments, rather than analyzing both reporters in parallel? Please clarify.

PR2. We will modify our analysis code and add error bars on Figure 4 in addition to the actual data points.

PR3. The reason to use the red channel is its lower background: the auto-fluorescence of yeast cells is lower in the red channel than in the green channel. We will add a supplementary figure showing this difference in background signal of negative cells.

Reviewer #2

1. In Figure 1, control experiments with no loxP sequences (i.e. original strain) should be performed to demonstrate specific binding of Cre/LiCre to loxP sequence.

PR4. We will perform an additional ChIP experiment, which will include the use of a negative-control strain with no *LoxP* sequence in the genome. Note that the original strain does not harbor the templates for qPCR amplicons P1, P2 and P3. It therefore cannot be used for comparisons. Since amplicon P2 matches a region of the mCherry coding sequence, we will use an unrelated mCherry strain for this control.

Reviewer #3

1. After completing the initial experiment, the authors discovered that their plasmids carry different numbers of V5 epitopes. I am wondering whether this was due to a recombination event happening during the experiment or whether the constructs were not sequence verified prior to use? In any case, an additional ChIP experiment using Cre and LiCre constructs with the identical number of tag-repeats will be necessary. The result, i.e. the strong reduction of DNA-binding of LiCre (which is close to the negative control), is quite remarkable given that LiCre is still

considerably active and high DNA affinities were observed in SPR experiments. In light of these counterindications, identical experiment conditions for test and reference group become even more important.

PR5. We will produce a Cre-V5 expression plasmid carrying 9 tag-repeats (instead of 12) and use it to perform an additional ChIP experiment where Cre and LiCre constructs have the identical number of tag-repeats.

2. The conclusion that DNA-binding of LiCre is completely light-independent is not entirely convincing to me. The differences between the light and dark conditions in Fig. 2d are indeed small, but the values for LiCre are almost on par with the vector control and therefore hard to interpret. Based on this experiment alone, one could even be inclined to argue that LiCre does not bind DNA at all (which is of course falsified by the later experiments), showing that the resolution of the corresponding dataset is too low to draw final conclusions. Light-independent DNA binding should either be confirmed by a more sensitive method or the conclusion statements on this matter should be revised accordingly.

PR6. We will revise the conclusion statements after completing the above-mentioned additional ChIP experiments (RP4 and RP5).

3. If I understand the explanations correctly, replicates and plotted data points refer to multiple samples (different colonies), that were handled in a single experiment, i.e. by one researcher at the same time/same day.

Yes, as listed in Supplementary Table S5, several datasets were produced, each at a different date. Each dataset includes one or more experimental conditions that were tested by one researcher across a series of experiments that were run over one or a few consecutive days. As explained in the Methods section, for each condition, three biological replicates were used where each replicate corresponded to a colony obtained from a different transformant (a transformant is a yeast clone identified on selective medium after introducing the LiCre-expression plasmid).

As already mentioned by the authors in the main text, this workflow explains the considerable differences between some of the results in the present manuscript and an identical experiment in a previous publication by the same authors. Providing truly independent experiments (performed on different days) that are therefore independent towards variables such as the fluctuation in incubation temperature (which was the issue in the described experiments) will be crucial, at least for the key datasets.

PR7. The datasets (Table S5) are indeed “truly independent” in the sense defined by the reviewer (performed on different days). To address the reviewer’s comment, we plan to describe the biological variability of the results across independent days when all variables (temperature, light, device, ...) are fixed. Datasets #3, #12 and #13 all include one common fixed set of variables: strain GY2517, period of 10 sec, duty cycle of 10%, illumination duration of 60 min, intensity 35 mW/cm², controlled temperature of 30°C, LED box device. We plan to add a supplementary figure to show rigorously the variability of the data across acquisition dates under this specific condition.

4. Fig. 2a-b depicts images relating to the purification procedure. These could be moved to the supplements as they don't provide any insight apart from the fact that the proteins were successfully purified.

PR8. We will move these panels to the supplements.

6. The difference between the ChIP results and the SPR results is striking but not mentioned in the discussion section. Also, the statement: "Finally, our results have practical implications on experimental protocols employing LiCre. First, given its high affinity for loxP (Fig. 5b), over-expressing LiCre at high levels will probably not increase its efficiency." (line 502) refers only to the affinity but seems to ignore the low DNA-occupancy of LiCre observed in Fig. 2d. Adapting the discussion section accordingly would improve the manuscript.

PR9. We will revise the discussion after completing the additional ChIP experiments of PR4 and PR5.

3. Description of the revisions that have already been incorporated in the transferred manuscript

Revisions appear in **yellow** in the transferred manuscript.

Reviewer #1

Line 110-115: Although described in the Methods section, a brief statement of dark and light treatment conditions would help readers better follow the experiments. Likewise, listing the three unrelated positions would improve the clarity.

The revised main text now mentions explicitly the illumination conditions.

Line 185: Is there a typo?

Yes, thank you, an “i” was missing and this has been fixed.

Line 236: The sentence "Importantly, we never observed recombination in the entire cell population" is ambiguous. I believe it means recombination was never observed in 100% of the cells.

Yes, this is what we meant. The revised text now writes “recombination in 100% of the cells”.

Line 245-249: The hypothesis of plasmid loss based on plating samples on selective and non-selective media without illumination assumes that loss of growth on selective media is only due to plasmid loss, without considering other factors like burden or toxicity. Moreover, the broad range of 10-30% makes it difficult to justify that the ~15% recombination-negative fraction falls within expected variation. The conclusion that LiCre-mediated recombination efficiency is close to 100% after prolonged photoactivation (Line 249, 301-303) is not fully convincing unless more evidence is provided.

We agree with the reviewer and this sentence was removed. The revised text now writes: “We hypothesized that this upper limit could, at least in part, result from the occasional loss of the LiCre-encoding plasmid. [...]. Thus, we cannot expect the proportion of cells undergoing LiCre-mediated recombination to exceed 70%-90% in our assay, even after prolonged illumination.”

Line 275-276: The authors suspect that the decrease in recombination efficiency at very high light intensity is possibly attributed to phototoxicity. Could photobleaching also contribute to this effect? A viability assay would help to validate the phototoxicity explanation.

We thank the reviewer for his/her careful reading. This sentence dates back from an earlier version of the manuscript, where the viability assay of Fig. 4b, which revealed no phototoxicity, was not achieved yet. We believe that, for a readership of biologists, the term “photobleaching” is usually associated with the loss of fluorescent emission after photo-induced damage of a dye or fluorescent protein. Since LiCre does not emit light, we’d like to avoid this term. The revised text now writes: “...possibly pointing to unknown inhibitory effects under prolonged or intense illumination. It is possible that high-intensity blue light causes damage to LiCre itself. It is also possible that, even if it does not compromise cell viability (Fig. 4b), intense illumination triggers a cell-stress response that inhibits recombination efficiency.”

Figure 1e: Please clarify whether the Western blots shown represent biological replicates.

Yes they do. The revised legend now writes: “(one biological replicate per lane)”.

For panel 4h and line 272, the statement that maximal activation was reached at 12 mW/cm² should be rephrased more cautiously, as no intermediate intensities between 12 and 35.6 mW/cm² were tested.

We rephrased this statement as “an overall increase of recombination efficiency with light intensity from 0 to 12 mW/cm².”

Reviewer #3

Minor comments:

1. *At the end of the Introduction, the authors mention that the interaction of the Cre heptamers was weakened via point mutations in LiCre. A short sentence about the engineering rationale behind this weakened interaction would help readers, who are not familiar with the author's prior work.*

We added the rationale in the revised text: “...by mutating residues E340 and D341 which, via their attraction to arginines R192 and R139, stabilize the association of the αN helix with the adjacent unit.”

Finally, we noted an error in the nucleotide sequence of primer 1R71 in Supp Table S3. This error was corrected.

4. Description of analyses that authors prefer not to carry out

Reviewer #1

Line 216: Have the authors considered performing surface plasmon resonance (SPR) to confirm the binding affinity of LiCre-V5 DNA?

We agree on the relevance of this suggestion but, regretfully, we cannot plan to run additional SPR experiments in the short term because i) the two authors of SPR acquisitions (A.D. and G.T.) are no longer available (they have left the laboratory and we currently do not have available staff with this expertise), and ii) our protein-production facility does not provide their services as they used to. We'd therefore need to achieve the production and purification of LiCre-V5 ourselves, or outsource it to another provider, which would imply long delays and uncertain production quality.

Line 345-346: While the model with x=2 provides a slightly better fit comparing to the others, the possibility of x=4 cannot be excluded. The inference that "photo-activation of at least two LiCre protomers enables recombination" is not sufficiently proven.

Lines 345-346 mention our model-based interpretation as a suggestion: “This suggests that the complex may not need four active LiCre units to be functional, and that photo-activation of only two units may be sufficient.” The sentence in lines 397-398 quoted by the reviewer is not a claim of mechanistic understanding but reports model adequacy: “the observed data are best explained by a model where the photo-activation of at least two LiCre protomers enables recombination”. It does not exclude the x=4 model: we used the “at least” formulation to exclude x=1, for which the fit is bad (non-realistic Koff), and to include x=2, 3, and 4 altogether. We nonetheless remain open to further rephrasing if the reviewer thinks this formulation is not appropriate.

Reviewer #2

2. *In Figure 2, the SPR experiments are robust and informative. However, the lack of measurement of DNA binding of light-activated LiCre is a notable gap, which will help understand whether the cooperativity of LiCre can be modulated by light. If it is difficult due to*

experimental conditions, there is lit- mimetic mutant of LOV2 (<https://www.nature.com/articles/nmeth.3926>).

We agree on the relevance of this suggestion and we regret that, as explained above, we cannot plan to run additional SPR experiments in the short term.

Reviewer #3

5. The kinetic characterization was only performed for LiCre. Especially for scientists, who have worked with wildtype Cre before, a side-by-side comparison with wt Cre would be valuable to judge the loss in reaction speed that has to be expected when switching from Cre to LiCre.

Unlike LiCre, wild-type Cre is constitutively active. Characterizing Cre's kinetics *in vivo* requires an induction or repression system. We observed that conditional expression systems, such as galactose-based induction, or methionine-based repression are problematic because i) the leaky (very low) expression levels are enough to induce a substantial degree of recombination and ii) the dynamics of expression induction or repression must then be taken into consideration, which would complicate and bring uncertainties (additional reaction steps and parameters) to the kinetic model. We therefore do not see how we could run a side-by-side comparison of the kinetics of LiCre and Cre *in vivo*.

Original submission

First decision letter

MS ID#: bio.062381

MS Title: Kinetic properties of optogenetic site-specific DNA recombination by LiCre-loxP

Authors: Alice Dufour; Hélène Duplus-Bottin; Thomas Boukeke-Lesplulier; Eliane Casassa; Gerard Triqueneaux; Leo Tarbouriech; Camille Darthenay-Kiennemann; Agnes Dumont; Catherine Moali; Daniel Jost; Gaël Yvert

I appreciate the effort put into completing several revisions and making a thorough plan for additional revisions that will strengthen the manuscript. After evaluating the planned and completed revisions for this manuscript, I **have decided to wait for resubmission of the completely revised version**, i.e., incorporating findings and text changes as indicated by PR1-PR9, before sending the manuscript back to the original reviewers.

At this stage, we also ask you to ensure your manuscript complies with our formatting guidelines. Provided you are able to fully address the referees' comments, we are positive about publication of your paper (we accept over 95% of revision submissions) and therefore hope you won't mind any extra work involved in reformatting your manuscript at this point.

Please upload both a 'clean' version of your Word file, along with a highlighted version clearly showing where you have made changes in the revised manuscript. Please avoid using 'Track changes' in Word files as these are lost in PDF conversion.

I should be grateful if you would also provide a point-by-point response detailing how you have dealt with the points raised by the reviewers in the 'Response to Reviewers' box. Please attend to all of the reviewers' comments. If you do not agree with any of their criticisms or suggestions please explain clearly why this is so.

First revisionAuthor responses to Reviewer comments**Manuscript number:** RC-2025-03141R**Corresponding author(s):** Gael Yvert

We thank the reviewers for their time and for their comments which helped us improve the quality of the manuscript. We present below a detailed point-by-point response to these comments. Revisions appear in **yellow** in the revised manuscript. Labels noted below as PR1, PR2, PR3 etc. correspond to planned revisions mentioned previously in our “revision plan”.

Reviewer #1

Line 110-115: Although described in the Methods section, a brief statement of dark and light treatment conditions would help readers better follow the experiments. Likewise, listing the three unrelated positions would improve the clarity.

The revised main text now mentions explicitly the illumination conditions and the three unrelated positions (lines 113-116).

Line 185: Is there a typo?

Yes, thank you, an “i” was missing and this has been fixed.

Line 216: Have the authors considered performing surface plasmon resonance (SPR) to confirm the binding affinity of LiCre-V5 DNA?

We agree on the relevance of this suggestion but, regretfully and as mentioned in our revision plan, we could not plan to run additional SPR experiments in the short term because i) the two authors of SPR acquisitions (A.D. and G.T.) are no longer available (they have left the laboratory and we currently do not have available staff with this expertise), and ii) our protein-production facility does not provide this service as they used to. We’d therefore need to achieve the production and purification of LiCre-V5 ourselves, or outsource it to another provider, which would imply long delays and uncertain production quality.

Line 233-234: To determine whether the observed difference in recombination efficiency is due to the genomic context of the reporter loci or due to the measurement accuracy of GFP and RFP signals, have the authors considered swapping the positions of GFP and RFP?

As planned (PR1), we compared strains with swapped positions. We observed that there are no difference in recombination efficiency between the two genomic positions (Revised Fig. 4f). This observation is mentioned in the revised main text (lines 241-247), and this additional dataset is now listed in the revised Supplementary Table S5.

Line 236: The sentence “Importantly, we never observed recombination in the entire cell population” is ambiguous. I believe it means recombination was never observed in 100% of the cells.

Yes, this is what we meant. The revised text now writes **“recombination in 100% of the cells”**.

Line 245-249: The hypothesis of plasmid loss based on plating samples on selective and non-selective media without illumination assumes that loss of growth on selective media is only due to plasmid loss, without considering other factors like burden or toxicity. Moreover, the broad range of 10-30% makes it difficult to justify that the ~15% recombination-negative fraction falls within expected variation. The conclusion that LiCre-mediated recombination efficiency is close

to 100% after prolonged photoactivation (Line 249, 301-303) is not fully convincing unless more evidence is provided.

We agree with the reviewer and this sentence was removed. The revised text now writes: “We hypothesized that this upper limit could, at least in part, result from the occasional loss of the LiCre-encoding plasmid.” [...]. “Thus, we cannot expect the proportion of cells undergoing LiCre-mediated recombination to exceed 70%-90% in our assay, even after prolonged illumination.”

Line 275-276: The authors suspect that the decrease in recombination efficiency at very high light intensity is possibly attributed to phototoxicity. Could photobleaching also contribute to this effect? A viability assay would help to validate the phototoxicity explanation.

We thank the reviewer for his/her careful reading. This sentence dates back from an earlier version of the manuscript, where the viability assay of Fig. 4b, which revealed no phototoxicity, was not achieved yet. We believe that, for a readership of biologists, the term “photobleaching” is usually associated with the loss of fluorescent emission after photo-induced damage of a dye or fluorescent protein. Since LiCre does not emit light, we’d like to avoid this term. The revised text now writes: “possibly pointing to unknown inhibitory effects under prolonged or intense illumination. It is possible that high-intensity blue light causes damage to LiCre itself. It is also possible that, even if it does not compromise cell viability (Fig. 4b), intense illumination triggers a cell-stress response that inhibits recombination efficiency.”

Line 345-346: While the model with $x=2$ provides a slightly better fit comparing to the others, the possibility of $x=4$ cannot be excluded. The inference that “photo-activation of at least two LiCre protomers enables recombination” is not sufficiently proven.

Lines initially numbered 345-346 mention our model-based interpretation as a suggestion: “This suggests that the complex may not need four active LiCre units to be functional, and that photo-activation of only two units may be sufficient.” The sentence in lines initially numbered 397-398 quoted by the reviewer is not a claim of mechanistic understanding but reports model adequacy: “the observed data are best explained by a model where the photo-activation of at least two LiCre protomers enables recombination”. It does not exclude the $x=4$ model: we used the “at least” formulation to exclude $x=1$, for which the fit is bad (non-realistic Koff), and to include $x=2$, 3, and 4 altogether. We nonetheless remain open to further rephrasing if the reviewer thinks this formulation is not appropriate.

Figure 1e: Please clarify whether the Western blots shown represent biological replicates.

Yes they do. The revised legend now writes: “(one biological replicate per lane)”.

Figure 4: Please include the error bars.

As planned (PR2), we added error bars on Figure 4 in addition to the actual data points.

Panel a - The authors integrated GFP and mCherry reporters at two different loci to avoid positional bias. Why then is only mCherry used as the ON readout in most experiments, rather than analyzing both reporters in parallel? Please clarify.

The reason to use the red channel is its lower background: the auto-fluorescence of yeast cells is lower in the red channel than in the green channel. As planned (PR3), we added Supplementary Figure S3 showing this difference in background signal of negative cells and the resulting higher single/noise ratio in the red channel. This difference in signal quality between the two channels is mentioned in the revised text.

For panel 4h and line 272, the statement that maximal activation was reached at 12 mW/cm^2 should be rephrased more cautiously, as no intermediate intensities between 12 and 35.6 mW/cm^2 were tested.

We rephrased this statement as “an overall increase of recombination efficiency with

light intensity from 0 to 12 mW/cm².”

Reviewer #2

1. In Figure 1, control experiments with no *loxP* sequences (i.e. original strain) should be performed to demonstrate specific binding of Cre/LiCre to *loxP* sequence.

As planned (PR4), we have performed an additional ChIP experiment, which included the use of a negative-control strain with no *LoxP* sequence in the genome. Note that the original strain does not harbor the templates for qPCR amplicons P1, P2 and P3. It therefore could not be used for comparisons. Since amplicons P2 & P3 match a region of the mCherry coding sequence, we used an unrelated mCherry strain for this control. Results are shown in revised Figure 1. The revised main text now reports our observation that “As expected, no ChIP signal was visible when using a control strain lacking the *loxP* site.”

In Figure 2, the SPR experiments are robust and informative. However, the lack of measurement of DNA binding of light-activated LiCre is a notable gap, which will help understand whether the cooperativity of LiCre can be modulated by light. If it is difficult due to experimental conditions, there is lit-mimetic mutant of LOV2 (<https://www.nature.com/articles/nmeth.3926>).

We agree on the relevance of this suggestion and we regret that, as explained in our revision plan, we could not plan to run additional SPR experiments in the short term.

Reviewer #3

1. After completing the initial experiment, the authors discovered that their plasmids carry different numbers of V5 epitopes. I am wondering whether this was due to a recombination event happening during the experiment or whether the constructs were not sequence verified prior to use? In any case, an additional ChIP experiment using Cre and LiCre constructs with the identical number of tag-repeats will be necessary. The result, i.e. the strong reduction of DNA-binding of LiCre (which is close to the negative control), is quite remarkable given that LiCre is still considerably active and high DNA affinities were observed in SPR experiments. In light of these counterindications, identical experiment conditions for test and reference group become even more important.

As planned (PR5), we have produced a Cre-V5 expression plasmid carrying 9 tag- repeats (instead of 12) and we used it to perform an additional ChIP experiment where Cre and LiCre constructs have an identical number of tag-repeats. The results, shown in revised Figure 1d, are very similar to the ones we observed previously. Results of initial Figure 1d-e are now presented in Supplementary Figure S1.

2. The conclusion that DNA-binding of LiCre is completely light-independent is not entirely convincing to me. The differences between the light and dark conditions in Fig. 2d are indeed small, but the values for LiCre are almost on par with the vector control and therefore hard to interpret. Based on this experiment alone, one could even be inclined to argue that LiCre does not bind DNA at all (which is of course falsified by the later experiments), showing that the resolution of the corresponding dataset is too low to draw final conclusions. Light-independent DNA binding should either be confirmed by a more sensitive method or the conclusion statements on this matter should be revised accordingly.

We performed the additional ChIP experiment described above, which confirmed that LiCre binds chromatin significantly as compared to the negative control (revised Fig. 1). The statistical significance (*P*-value = 0.01) was the same as in the previous - independent - experiment (Supplementary Figure S1). The resolution of the combined datasets is therefore sufficient to conclude that, although weak, LiCre binding to chromatin is reliably above background levels. The revised text now writes that this binding “reproducibly reached statistical significance”.

Regarding the effect of light, we - again - observed a mild increase of the ChIP signal (light relative to dark condition) for probe P2 which, this time, reached significance at P -value = 0.05 (revised Figure 1). To account for this modest effect, we revised the main text as follows: “illumination prior to chromatin immuno-precipitation had **only a very small** effect on LiCre ChIP signals. **This improvement was marginally significant in one experiment (Fig. 1d, probe P2) and not significant at all in another independent experiment (Supplementary Fig. S1)**. We conclude that, *in vivo*, LiCre **weakly** binds *loxP*-containing chromatin in the dark and that this binding **may be slightly improved** in light conditions.”

These text revisions correspond to our planned revision PR6.

3. *If I understand the explanations correctly, replicates and plotted data points refer to multiple samples (different colonies), that were handled in a single experiment, i.e. by one researcher at the same time/same day.*

Yes, as listed in Supplementary Table S5, several datasets were produced, each at a different date. Each dataset includes one or more experimental conditions that were tested by one researcher across a series of experiments that were run over one or a few consecutive days. As explained in the Methods section, for each condition, three biological replicates were used where each replicate corresponded to a colony obtained from a different transformant (a transformant is a yeast clone identified on selective medium after introducing the LiCre-expression plasmid).

As already mentioned by the authors in the main text, this workflow explains the considerable differences between some of the results in the present manuscript and an identical experiment in a previous publication by the same authors. Providing truly independent experiments (performed on different days) that are therefore independent towards variables such as the fluctuation in incubation temperature (which was the issue in the described experiments) will be crucial, at least for the key datasets.

The datasets (Table S5) are indeed “truly independent” in the sense defined by the reviewer (performed on different days). As planned (PR7), we visualized the biological variability of the results across independent days when all variables (temperature, light, device, ...) are fixed. Datasets #3, #12 and #13 all include one common fixed set of variables: strain GY2517, period of 10 sec, duty cycle of 10%, illumination duration of 60 min, intensity 35 mW/cm², controlled temperature of 30 °C, LED box device. We added Supplementary Figure S4 showing the variability of the data across acquisition dates under this specific condition. The revised main text now writes “We therefore conducted subsequent experiments under strict temperature-controlled conditions (30 °C), **which ensured reproducibility across independent datasets (Supplementary Fig. S4).**” (line 271).

Minor comments:

1. *At the end of the Introduction, the authors mention that the interaction of the Cre heptamers was weakened via point mutations in LiCre. A short sentence about the engineering rationale behind this weakened interaction would help readers, who are not familiar with the author's prior work.*

We added the rationale in the revised text: “...by mutating **residues E340 and D341** which, via their attraction to arginines R192 and R139, stabilize the association of the α N helix with the adjacent unit;”

4. *Fig. 2a-b depicts images relating to the purification procedure. These could be moved to the supplements as they don't provide any insight apart from the fact that the proteins were successfully purified.*

As planned (PR8), we moved these panels to Supplementary Fig. S2.

5. *The kinetic characterization was only performed for LiCre. Especially for scientists, who have worked with wildtype Cre before, a side-by-side comparison with wt Cre would be valuable to judge the loss in reaction speed that has to be expected when switching from Cre*

to LiCre.

Unlike LiCre, wild-type Cre is constitutively active. Characterizing Cre's kinetics *in vivo* requires an induction or repression system. We observed that conditional expression systems, such as galactose-based induction, or methionine-based repression are problematic because i) the leaky (very low) expression levels are enough to induce a substantial degree of recombination and ii) the dynamics of expression induction or repression must then be taken into consideration, which would complicate and bring uncertainties (additional reaction steps and parameters) to the kinetic model. As explained in our revision plan, we therefore do not see how we could run a side-by-side comparison of the kinetics of LiCre and Cre *in vivo*.

6. *The difference between the ChIP results and the SPR results is striking but not mentioned in the discussion section. Also, the statement: "Finally, our results have practical implications on experimental protocols employing LiCre. First, given its high affinity for loxP (Fig. 5b), over-expressing LiCre at high levels will probably not increase its efficiency." (line 502) refers only to the affinity but seems to ignore the low DNA-occupancy of LiCre observed in Fig. 2d. Adapting the discussion section accordingly would improve the manuscript.*

This point is now specifically discussed in a section starting with "Our *in vivo* and *in vitro* analyses of LiCre:DNA binding provided contrasting results..." (lines 415-427). This text revision corresponds to our planned revision PR9.

ADDITIONAL REVISION

Finally, we noted an error in the `nucleotidesequenceofprimer1R71inSuppTableS3`. This error was corrected.

Second decision letter

MS ID#: bio.062381R1

MS Title: Kinetic properties of optogenetic site-specific DNA recombination by LiCre-loxP

Authors: Alice Dufour; Hélène Duplus-Bottin; Thomas Boukeke-Lesplulier; Eliane Casassa; Gerard Triqueneaux; Leo Tarbouriech; Camille Darthenay-Kiennemann; Agnes Dumont; Catherine Moali; Daniel Jost; Gaël Yvert

I am happy to tell you that your manuscript has been accepted for publication in Biology Open, pending our standard publication integrity checks. It was accepted on 3rd March 2026.

Reviewer 1

Comments for the author

In their revision, the authors performed multiple additional experiments to clarify the uncertainties mentioned by the other reviewers and me. They have convincingly addressed all my remarks and comments.

I recommend the publication of the manuscript without further requests.

Congratulations to the authors!

Reviewer 2

Comments for the author

The authors have addressed all my concerns.

Reviewer's Responses to Questions

Experimental quality

Does each figure have the proper controls?

If 'No', please indicate reasons in Comments for Author box below.

Reviewer #1:

- Yes

Reviewer #2:

- Yes

Were the data analyzed using appropriate statistical tests?

If 'No', please indicate reasons in Comments for Author box below.

Reviewer #1:

- Yes

Reviewer #2:

- Yes

Reproducibility

Were experiments performed using adequate number of biological replicates?

If 'No', please indicate reasons in Comments for Author box below.

Reviewer #1:

- Yes

Reviewer #2:

- Yes

Does the methods section provide sufficient detail to permit reproducibility?

If 'No', please indicate reasons in Comments for Author box below.

Reviewer #1:

- Yes

Reviewer #2:

- Yes

Completeness

Are the manuscript's conclusions supported by the data?

If 'No', please indicate reasons in Comments for Author box below.

Reviewer #1:

- Yes

Reviewer #2:

- Yes

Scholarship

Do the authors cite and discuss the merits of data that would argue for and against their conclusion?

If 'No', please indicate reasons in Comments for Author box below.

Reviewer #1:

- Yes

Reviewer #2:

- Yes

Does the manuscript title & abstract accurately reflect the contents of the manuscript, without hyperbole?

If 'No', please indicate reasons in Comments for Author box below.

Reviewer #1:

- Yes

Reviewer #2:

- Yes